# Protease-activation using anti-idiotypic masks enables tumor specificity of a folate receptor 1-T cell bispecific antibody

Martina Geiger [1,2], Kay-Gunnar Stubenrauch[3], Johannes Sam [1], Wolfgang F. Richter [4], Gregor Jordan[3], Jan Eckmann[3], Carina Hage [3], Valeria Nicolini[1], Anne Freimoser-Grundschober [1], Mirko Ritter[5], Matthias E. Lauer [4], Henning Stahlberg [6], Philippe Ringler[6], Jigar Patel[7,10], Eric Sullivan[7,10], Sandra Grau-Richards[1], Stefan Endres[2,8,9], Sebastian Kobold [2,8,9], Pablo Umaña[1], Peter Brünker[1,11] & Christian Klein [1,11 ✉]

T-cell bispecific antibodies (TCBs) crosslink tumor and T-cells to induce tumor cell killing. While TCBs are very potent, on-target off-tumor toxicity remains a challenge when selecting targets. Here, we describe a protease-activated anti-folate receptor 1 TCB (Prot-FOLR1-TCB) equipped with an anti-idiotypic anti-CD3 mask connected to the anti-CD3 Fab through a tumor protease-cleavable linker. The potency of this Prot- FOLR1-TCB is recovered following protease-cleavage of the linker releasing the anti-idiotypic anti-CD3 scFv. In vivo, the Prot-FOLR1-TCB mediates antitumor efficacy comparable to the parental FOLR1-TCB whereas a noncleavable control Prot-FOLR1-TCB is inactive. In contrast, killing of bronchial epithelial and renal cortical cells with low FOLR1 expression is prevented compared to the parental FOLR1-TCB. The findings are confirmed for mesothelin as alternative tumor antigen. Thus, masking the anti-CD3 Fab fragment with an anti-idiotypic mask and cleavage of the mask by tumor-specific proteases can be applied to enhance specificity and safety of TCBs.

[1] Roche Pharma Research & Early Development, Roche Innovation Center Zurich, Wagistrasse 10, 8952 Schlieren, Switzerland. [2] Center of Integrated Protein Science Munich (CIPS-M) and Division of Clinical Pharmacology, Department of Medicine IV, Klinikum der Universität München, Lindwurmstraße 2a, Member of the German Center for Lung Research (DZL), 80337 Munich, Germany. [3] Roche Pharma Research & Early Development, Roche Innovation Center Munich, Nonnenwald 2, 82372 Penzberg, Germany. [4] Roche Pharma Research & Early Development, Roche Innovation Center Basel, Grenzacherstrasse 124, 4070 Basel, Switzerland. [5] Roche Diagnostics, CPS Research and Development, Nonnenwald 2, 82372 Penzberg, Germany. [6] Center for Cellular Imaging and Nano Analytics, Biozentrum, University of Basel, 4070 Basel, Switzerland. [7] Roche Sequencing, NimbleGen, Madison, WI 53719, USA. [8] Einheit für Klinische Pharmakologie (EKLiP), Helmholtz Zentrum München, German Research Center for Environmental Health (HMGU), Neuherberg, Germany. [9] German Center for Translational Cancer Research (DKTK), Partner Site Munich, Munich, Germany. [10]Present address: Nimble Therapeutics Inc., 500S Rosa Rd, Madison, WI 53719, USA. [11]These authors contributed equally: Peter Brünker, Christian Klein. ✉email: christian.klein.ck1@roche.com

Cancer immunotherapy proves clinical efficacy in several indications[1]. T-cell bispecific antibodies (TCBs) are antibodies targeting an antigen expressed on target cells and the CD3ε subunit of the T-cell receptor on T cells to mediate tumor cell lysis. We recently described 2 + 1 TCBs consisting of an inert Fc region, two tumor antigen-binding Fab fragments and one Fab fragment binding to CD3 on the T-cell receptor[2,3]. The addition of the Fc part, compared to smaller antibody formats[4], increases the half-life while systemic activation of immune cells via FcγR or complementary binding is prevented by introduction of P329G LALA Fc mutations[5]. When T- and tumor cells are simultaneously bound by the TCB, this results in subsequent T-cell activation and potent serial tumor cell killing. Recently, efficacy of a carcinoembryonic antigen (CEA)-specific CEA-TCB (RG7802) was demonstrated[2,3]. CEA-TCB efficiently kills tumor cells with high CEA expression while sparing normal cells with low CEA expression. The threshold of T-cell activation is >10,000 CEA molecules per cell for efficient killing. However, for other TCBs like the folate receptor 1 (FOLR1, FolRα) TCB (Griessinger, #1759) described below such a threshold does not exist, and related molecules like ImmTacs can kill cells with low target expression in the range of several hundred receptors as recently demonstrated for peptide MHC complexes as target[6]. Thus, physiological tissue expression of a given antigen can be critical when developing TCBs or other T-cell activating therapies such as CAR-T cells[7]. Improving the specificity of TCBs would increase the number of potential tumor targets. Proteases like serine proteases (e.g. matriptase), cysteine proteases (e.g. cathepsin S) and matrix metalloproteinases (e.g. MMP-2 and MMP-9) are overexpressed in several cancer types[8]. Matriptase, matrix metalloproteinase 2 (MMP-2, gelatinase A) and matrix metalloproteinase 9 (MMP-9, gelatinase B) are overexpressed e.g. in breast- and ovarian carcinoma[9–19]. MMP-2 and MMP-9 activity was detected in cervical, breast and ovarian carcinoma and ascites of patients with epithelial ovarian cancer (EOC) but not in the serum of these patients[20]. While matriptase can be detected in normal epithelial cells, matriptase activity is mainly detected in cancer[21]. Therefore, these proteases are suitable as cancer-specific

activators of potent agents like TCBs allowing the targeting of otherwise unsuitable antigens.

We have previously generated an FOLR1-TCB (Griessinger, #1759 4, shift the rest). FOLR1 is overexpressed in various tumors including ovarian, lung and breast cancer[22], but is also expressed to lower degrees on normal cells e.g. in the lung and kidney[23]. While FOLR1-TCB was efficacious in vitro and in xenograft models, severe on-target toxicity in the lung of non-human primates was observed[24]. Based on this experience, we chose FOLR1-TCB as a relevant model to show proof-of-concept for masking the anti-CD3 moiety with an anti-idiotypic antibody scFv fused via a protease cleavable linker to the TCB. For this purpose, we fused a specific anti-idiotypic anti-CD3 scFv N-terminally to the anti-CD3 variable heavy chain connected by a protease cleavable linker and demonstrated that active proteases located in the tumor microenvironment lead to cleavage and subsequent unmasking of the anti-CD3 targeting moiety. Unmasking results in efficient killing of FOLR1-positive tumor cells in vitro and in vivo while sparing normal cells with low FOLR1 expression.

## Results

**Engineering of protease-activated antibodies.** In non-human primates, on-target toxicity has been observed when a highly potent FOLR1-TCB (based on clone 16D5) with EC50 values in the single-digit pM range was evaluated in tolerability experiments at single doses as low as 10 μg/kg[24]. To overcome this limitation, our approach was to block CD3 binding with an anti-idiotypic anti-CD3 scFv that can be cleaved off by tumor-specific proteases. The Prot-FOLR1-TCB is supposed to be specifically activated in the tumor microenvironment releasing the blocking anti-idiotypic CD3 scFv (Fig. 1). To prove the feasibility of blocking the anti-CD3 Fab, we first engineered a monovalent anti-CD3 antibody with a N-terminally fused anti-idiotypic anti-CD3 scFv (Fig. 2a). To test the concept of protease-activation in the TCB format, we then engineered a Prot-FOLR1-TCB by fusion of this anti-idiotypic anti-CD3 scFv to the anti-CD3 Fab (Fig. 2b).

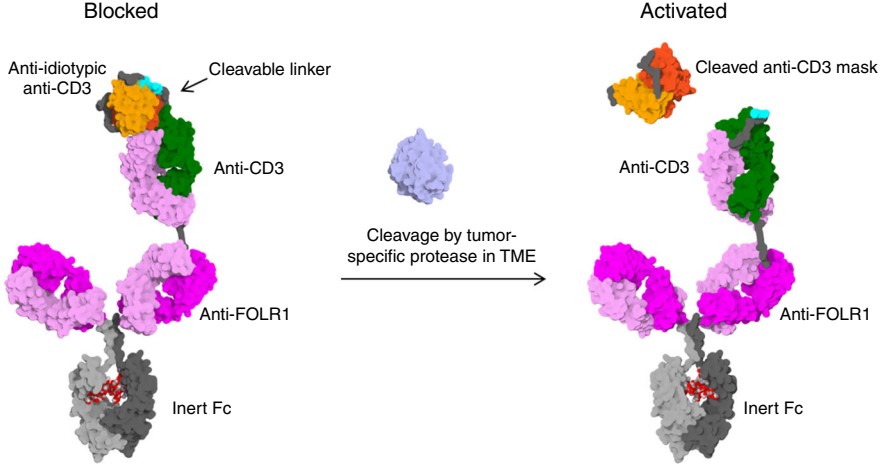

**Fig. 1 Mode of action of protease-activated FOLR1-TCB.** On the left panel, the anti-CD3 Fab is blocked by an anti-idiotypic anti-CD3 scFv and thus cannot activate T cells against FOLR1-expressing cells. On the right panel, the linker containing a tumor-specific protease site has been cleaved and the anti-CD3 moiety is active leading to lysis of FOLR1-expressing cells. The figure shows an idealized representation of the Prot-FOLR1-TCB before (left) and after (right) cleavage at the matriptase cleavage site (cyan). The model is based on the full-length IgG crystal structure with PDB ID 1hzh. The protected anti-CD3ε Fab was modeled based on the crystal structure of an idiotype-anti-idiotype Fab complex structure with PDB ID 1iai. The catalytic domain of matriptase (crystal structure with PDB ID 1eax) is shown for reference (gray). Visualized with Biovia Discovery Studio 17R2 and arranged with GIMP. TME tumor microenvironment.

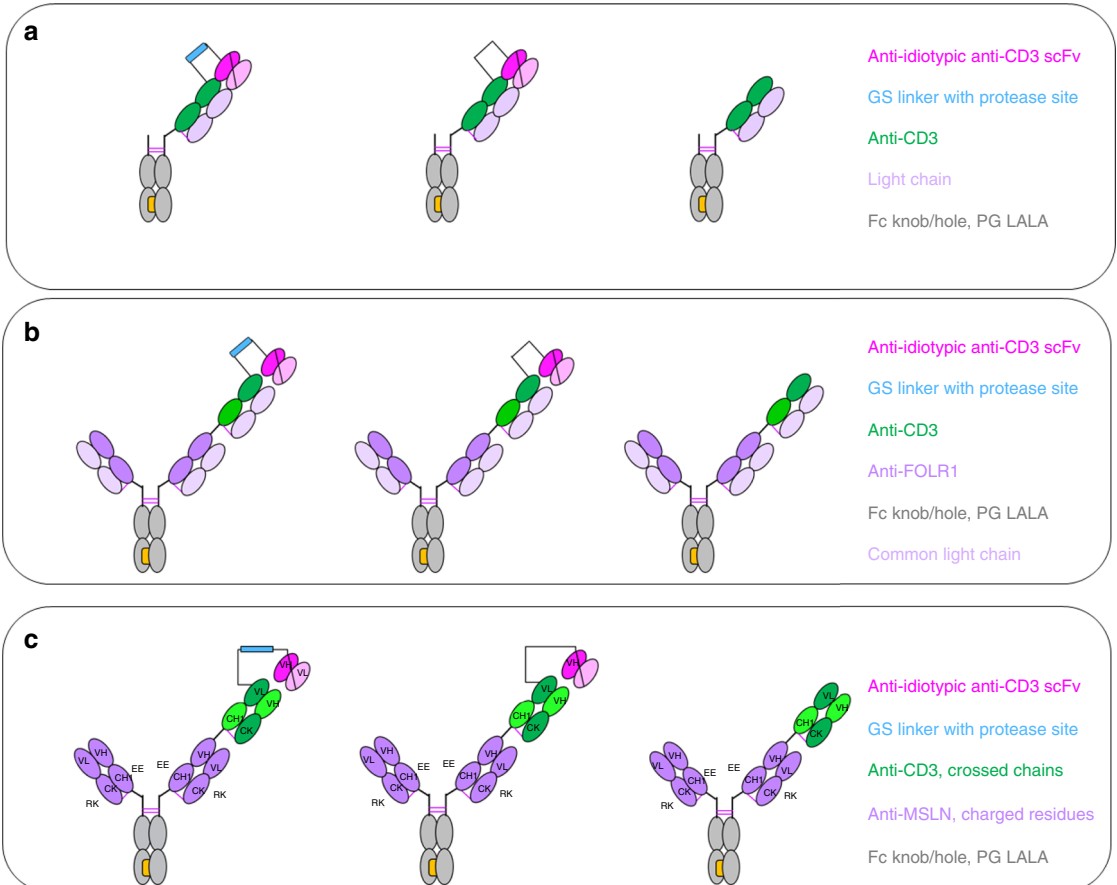

**Fig. 2 Design of protease-activated anti-CD3 in IgG and TCB format.** "Knobs-into-holes" technology was used for the generation of heterodimeric molecules and PG LALA mutations were inserted to prevent FcγR binding. **a** Protease-activated monovalent anti-CD3 IgG (Prot-mαCD3 IgG) with N-terminal fusion of anti-idiotypic anti-CD3 scFv (disulfide-stabilized) and linker containing a protease cleavage site, monovalent anti-CD3 IgG with an anti-idiotypic anti-CD3 scFv N-terminally fused by a noncleavable glycine-serine linker and monovalent anti-CD3 IgG. **b** Prot-FOLR1-TCB with N-terminal fused anti-idiotypic anti-CD3 scFv and linker containing a protease cleavage site, Prot-FOLR1-TCB with N-terminal fused anti-idiotypic anti-CD3 scFv and a noncleavable GS linker and FOLR1-TCB. **c** Prot-MSLN-TCB with N-terminal fused anti-idiotypic anti-CD3 scFv and linker containing a protease cleavage site, Prot-MSLN-TCB with N-terminal fused anti-idiotypic anti-CD3 scFv and noncleavable GS linker and MSLN-TCB. For FOLR1 and CD3 binders, a common light chain could be used. For MSLN, the CrossMab technology was used to foster correct pairing of chains.

The anti-CD3 Fab is a humanized anti-CD3ε Fab with monovalent affinity to human/cyno CD3ε in single-digit nM range. Anti-idiotypic anti-CD3 antibodies were identified from hybridomas from mice immunized with anti-CD3 F(ab')₂ fragments. Several anti-idiotypic anti-CD3 IgGs were screened for their avidity to anti-CD3 F(ab')2 and tested in IgG and TCB format for masking efficiency. From these, the two anti-idiotypic anti-CD3 masks 4.15 and 4.32 were chosen to be tested in the Prot-FOLR1-TCB format. While the affinity to the anti-CD3 Fab was the same (~2 nM), the affinity to the anti-CD3 Fab in the TCB format was 20 nM for the clone 4.15 and 10 nM for the clone 4.32 (Supplementary Table 1). The lower affinity of the masking scFv resulted in lower masking-efficiency shown in target cell killing for target cells with very high FOLR1 expression (Supplementary Fig. 1A, B). As both masks were released resulting in comparable potency of activated Prot-FOLR1-TCB, we chose the 4.32 mask for further characterization (Supplementary Fig. 1A, B).

The Prot-FOLR1-TCB was generated by fusing the anti-idiotypic anti-CD3 scFv to the variable heavy (VH) chain of the anti-CD3 Fab fragment (Fig. 2b, c). An additional disulfide bridge (VH44-VL100) was inserted to increase stability and reduce aggregate formation of IgG-scFv fusion molecules[24–27]. The variable heavy and the variable light chain of the scFvs were connected by a (G₄S)₄ linker. For the Prot-FOLR1-TCB (clone 16D5), a common light chain for both the FOLR1 and anti-CD3 Fab fragment was applied, facilitating correct light chain association and "knobs-into-hole" technology to enable correct Fc heterodimerization[28,29]. All antibodies carried an inert Fc with P329G LALA5 mutations. Fusion of the anti-idiotypic anti-CD3 scFv was achieved via a synthetic linker sequence (33 amino acids) comprising different protease cleavage sites (Supplementary Table 2). FOLR1-TCB without a blocking moiety or a respective Prot-FOLR1-TCB with a noncleavable linker formed by a (G₄S)₃−(G₇S)₁−(G₄S)₂ linker were used as controls (Supplementary Table 2). An analogous protease-activated mesothelin TCB (Prot-MSLN-TCB) was generated using a CrossMAb^VH-VL format[29,30] with charged residues[31] in the Fab fragments to enable correct light chain pairing based on the humanized Mesothelin antibody SS1 (Fig. 2c).

All antibodies were transiently produced in HEK293 cells, purified and analyzed for integrity and monomer content (Supplementary Table 3). The antibodies were stable once produced and the linkers, containing different cleavage sites, were stable during the purification process. The protease-activated IgGs and TCBs containing either a matriptase (matA site) or a MMP2, -9-matriptase (MMP-matA site) cleavage site were additionally analyzed for stability and cleavage with

recombinant human matriptase by capillary electrophoresis. The antibodies were stable at 4 °C and at 37 °C for a minimum of 48 h and were cleaved in vitro by recombinant human matriptase (Supplementary Fig. 2). In order to demonstrate the cleavage site specificity, the constructs were incubated either with recombinant human matriptase (rhMatriptase), recombinant human MMP-2 (rhMMP-2) or recombinant human MMP-9 (rhMMP-9). The cleavage of the linker was indirectly measured in a cell assay via luminescence, indicating activation of Jurkat NFAT-cells upon CD3 binding. The MMP site used herein was cleaved by rhMMP-2 and rhMMP-9, whereas the matA cleavage site was cleaved by rhMatriptase (Supplementary Fig. 3). In order to further demonstrate the thermal stability of the protease linkers, different antibodies, containing an MMP or a combined MMP-matA cleavage site, were heated up to 85 °C and analyzed for aggregate formation and melting temperature. All antibodies tested were stable up to 58 °C (Supplementary Table 4). Additionally, no cleavage was detectable for the Prot-FOLR1-TCB containing matA or MMP-matA site after incubation in human serum for 14 days at 37 °C (Supplementary Fig. 4).

**Prot-mαCD3 IgG can bind to CD3ε antigen after activation**. To further characterize the antibodies and prove feasibility of re-activation of the anti-CD3 Fab fragment, we verified the globular integrity and the functional activity of the protease-activated monovalent anti-CD3 IgG (Prot-mαCD3 IgG) by negative stain transmission electron microscopy (NS-TEM) and atomic force microscopy (AFM). Class-averages derived from multivariate statistical analysis of particles in micrographs recorded with NS-TEM confirmed the expected globular structure of the Prot-mαCD3 IgG (Fig. 3a). The resolution achieved with this method allowed to distinguish between three globular fragments. The C-terminal domain of the triangularly shaped Fc-fragment was linked to an anti-CD3 Fab fragment which was characterized by a central hole. The distal N-terminal domain resembled the anti-idiotypic anti-CD3 scFv masking moiety and had a size expected for VH/VL domains, whereas molecules lacking this moiety were shorter. Molecule complexed with CD3-Fc antigen fusion were larger and characterized by two Fc fragments at the distal ends. Unfortunately, the achieved given resolution was not sufficient to annotate them (Fig. 3a).

The unmasking functionality of the construct was structurally confirmed on the level of single molecules using tapping-mode AFM in liquid. Masked Prot-mαCD3 molecules were trapped on a surface of mica and monitored for structural changes introduced by matriptase treatment and the addition of the Fc-CD3εδ ligand. Topology and size changes were observed and attributed to the unmasking-complexation process (Fig. 3b and Supplementary Fig. 5). The changes match the molecule structures observed with NS-TEM with respect to topology and length,. Thus, at the beginning of the experiment, the molecules resembled a chain of three jointed segments. The treatment with matriptase resulted in a fraction of shorter molecules with only two segments, and the treatment with the more bulky Fc-CD3εδ ligand finally resulted in the expected elongated complex over time.

**Masking of anti-CD3 Fab impairs CD3 binding on T cells**. To prove that crosslinking of the monovalent anti-CD3 binder is required for successful T-cell activation, we compared the mαCD3 IgG both in the presence or absence of plate-coated anti-human Fc antibody, using Jurkat NFAT-cells or peripheral blood mononuclear cells (PBMCs) as effector cells (Fig. 4). The Jurkat NFAT reporter cell line contained a nuclear factor of activated T-cell (NFAT) promoter upstream of a luciferase gene. Binding and

subsequent crosslinking of CD3ε induces downstream signaling resulting in luciferase expression which was quantified via luminescence after substrate addition. In addition, activation of natural T cells in the PBMC fraction was analyzed by FACS using the T-cell activation marker CD69. We clearly detected dose-dependent Jurkat NFAT and CD8 T-cell activation only in the presence of plate-coated anti-human Fc antibody, indicating that this activation was dependent on crosslinking of the monovalent CD3 antibody. In contrast, no activation was detected in the absence of anti-human Fc antibody coated on plates (Fig. 4a, b). In subsequent assays we used the Jurkat NFAT reporter assay or human PBMCs to analyze CD3 binding of Prot-mαCD3 IgG (masked, cleaved or noncleavable) compared to mαCD3 IgG. The Prot-mαCD3 IgG, containing an matA site, was activated by cleavage with rhMatriptase to show full activity (Fig. 4c, d). In addition, blocking of CD3 binding was clearly dependent on the masking by anti-idiotypic CD3 scFvs as the N-terminal fusion of an unrelated Fab did not block CD3 binding (Fig. 4c, d).

**Masking-efficiency depends on antigen expression level**. In order to demonstrate the reduction of target cell lysis by blocking of the anti-CD3 Fab, the dose-dependent T-cell killing of FOLR1-positive tumor cells by Prot-FOLR1-TCBs was measured after 48 h. Masking-efficiency was investigated for HeLa and Skov-3 cells that express high (approx. 2 mio antigen-binding sites (ABS)/cell) or medium (approx. 0.1 mio ABS/cell) levels of FOLR1 antigen, respectively. Prot-FOLR1-TCB, precleaved with rhMatriptase, showed dose-dependent killing for both cell lines (Fig. 5a). The Prot-FOLR1-TCB containing a noncleavable site did not induce any killing for Skov-3 cells in the indicated concentration range. For HeLa cells with a very high FOLR1 expression, the EC50 of the Prot-FOLR1-TCB (noncleavable site) was significantly reduced up to 4000-fold compared to the cleaved TCB (Fig. 5a). The activity of Prot-FOLR1-TCB was restored after linker cleavage.

Several proteases are described to be overexpressed in ovarian carcinoma. We chose matrix metalloproteinase-2,-9 (MMP-2, MMP-9) and matriptase to compare cleavage by cellular proteases. We compared target cell cytotoxicity for Prot-FOLR1-TCB containing either an MMP site, a matriptase (matA) site or a combined MMP-matA site for cleavage by naturally expressed proteases (Supplementary Fig. 6A, B). Dose-dependent killing assays using FOLR1-positive HeLa and Skov-3 cells revealed a higher potency for the Prot-FOLR1-TCB with the combined MMP-matA site (EC50 approx. 6- to 7-fold lower for HeLa cells, EC50 approx. 2−3-fold lower for Skov-3 cells) (Supplementary Fig. 6A, B). For this reason the MMP-matA site was chosen for further evaluation. To analyze if the potency of the Prot-FOLR1-TCB can be recovered after linker cleavage, we compared FOLR1-TCB, Prot-FOLR1-TCB (precleaved with rhMatriptase) and Prot-FOLR1-TCB for dose-dependent target cell cytotoxicity and Granzyme B release using HeLa and Skov-3 cells (Fig. 5c–f). The Prot-FOLR1-TCB was cleaved by cellular proteases and its potency was comparable to the precleaved Prot-FOLR1-TCB and the FOLR1-TCB using HeLa cells with high FOLR1 expression. For Skov-3 cells with medium FOLR1 expression the potency of the Prot-FOLR1-TCB was lower compared to the precleaved Prot-FOLR1-TCB and the FOLR1-TCB (Fig. 5c–f). However, the potency of the fully activated (precleaved) Prot-FOLR1-TCB was comparable to FOLR1-TCB for both cell lines (Fig. 5c–g). The Prot-FOLR1-TCB with a glycine and serine (GS) noncleavable site induced significantly less T-cell-mediated cytotoxicity for HeLa and Skov-3 cells compared to the Prot-FOLR1-TCB containing an MMP-matA site. At the highest concentration (10 nM), a maximal

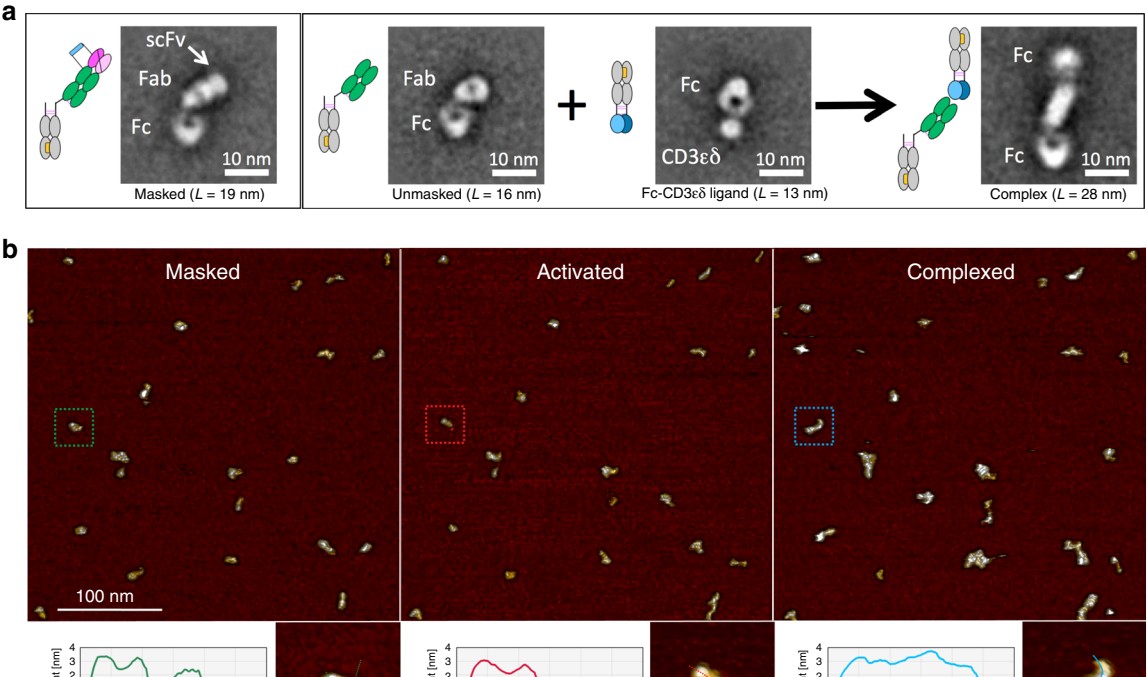

**Fig. 3 Masking of CD3ε binder reversibly impairs binding to CD3ε antigen.** Single particle analysis of Prot-mαCD3 IgG. **a** Negative stain transmission electron microscopy (NS-TEM) of the Prot-mαCD3 IgG containing a matriptase cleavable linker. 2D class-averages, as they result from multivariate statistical analysis of raw micrographs, are displayed in boxes (box size 36 nm). The Fc parts have triangular shapes with a central hole and Fabs are more elongated. At the distal end of the Fab the protective scFv appears as a single small additional density (see white arrow). The complex formation between antibody and ligand is identified by the presence of two Fc-regions linked together by an elongated structure corresponding to Fab and bound ligand. Due to the flexibility of the complex, one of the Fc displays less details in most of the class-averages. **b** Time-lapsed tapping-mode AFM data of individual Prot-mαCD3 IgG molecules have been recorded before (left), during activation with matriptase (middle), as well as after complexation with CD3 antigen (right) in a qualitative manner. The process is accompanied by length and shape changes which are noticed in the morphology maps. The treatment of Prot-mαCD3 (left) with matriptase results in an activated and shorter molecule (middle). The complexation of the activated molecules with Fc-CD3 gives the expected elongated and larger molecule (right). The lengths changes are compared with cross-section profiles starting with the individual masked molecule (green) undergoing activation (red) and after been complexed (blue), the profiles match the particles dimensions measured with NS-TEM, as depicted in (**a**). Representative AFM images of *n* = 3 experiments shown.

T-cell-mediated cytotoxicity of 20% was observed for highly FOLR1-positive HeLa cells (Fig. 5c). Quantification of cytotoxic granule granzyme B after incubation of target cells with PBMCs and Prot-FOLR1-TCB revealed a dose-dependent TCB-mediated release of granzyme B. The granzyme B release mediated by the activated Prot-FOLR1-TCB was comparable to the FOLR1-TCB whereas no granzyme B release could be detected for the masked Prot-FOLR1-TCB with a noncleavable linker (Fig. 5e, f).

In order to analyze the kinetics of T-cell-mediated cytotoxicity mediated by the Prot-FOLR1-TCB, we investigated tumor cell growth for MDA-MB-231 NucLight red cancer cells (medium FOLR1 expression) during coincubation with PBMCs and the Prot-FOLR1-TCB. We detected significant growth inhibition for the Prot-FOLR1-TCB with the MMP-matA site compared to the Prot-FOLR1-TCB containing a noncleavable site (Fig. 5g).

**The concept is applicable for other TCBs.** A major advantage of masking the anti-CD3 Fab fragment is that the concept can be applied for TCBs targeting different tumor antigens. For proof-of-concept we analyzed dose-dependent T-cell-mediated cytotoxicity on mesothelin (MSLN)-positive tumor cell lines NCIH596 (approx. 80,000 MSLN ABS/cell) and AsPC1 (approx. 50,000 MSLN ABS/cell) mediated by an analogous Prot-MSLN-TCB. The precleaved Prot-MSLN-TCB and the MSLN-TCB were comparable regarding their potency for both cell lines (Supplementary

Fig. 6c, d). The Prot-MSLN-TCB, cleaved by proteases expressed only by the target cell line, was comparable to the MSLN-TCB for NCI H596 cells (Supplementary Fig. 6C), whereas for AsPC-1 cells (Supplementary Fig. 6D), the in vitro activated Prot-MSLN-TCB did not reach the potency of the MSLN-TCB. The masked MSLN-TCB containing a GS noncleavabe linker prevented T-cell-mediated killing in the indicated concentration range for both cell lines.

**Cytotoxicity of FOLR1-TCB is abolished by blocking anti-CD3.** As FOLR1 is known to be expressed to a low extent by normal lung and kidney cells[23], we tested the reduction of potential on-target toxicity by analyzing target cell killing of Prot-FOLR1-TCB on primary human bronchial epithelial cells (HBEpiC, <1000 FOLR1 ABS/cell) and primary human renal cortical epithelial cells (HrcEpiC, <1000 FOLR1 ABS/cell). Neither killing nor T-cell activation (CD69 increase for CD8 T cells) was observed for HBEpiC and HrcEpiC cells using Prot-FOLR1-TCB and huPBMCs (Fig. 6a–d). In contrast, the parental FOLR1-TCB induced cell lysis and T-cell activation for both cell types (Fig. 6a–d).

**Prot-FOLR1-TCB can be activated by cancer explants.** In order to show that the Prot-FOLR1-TCB can be activated by human patient-derived samples expressing FOLR1 as a tumor target

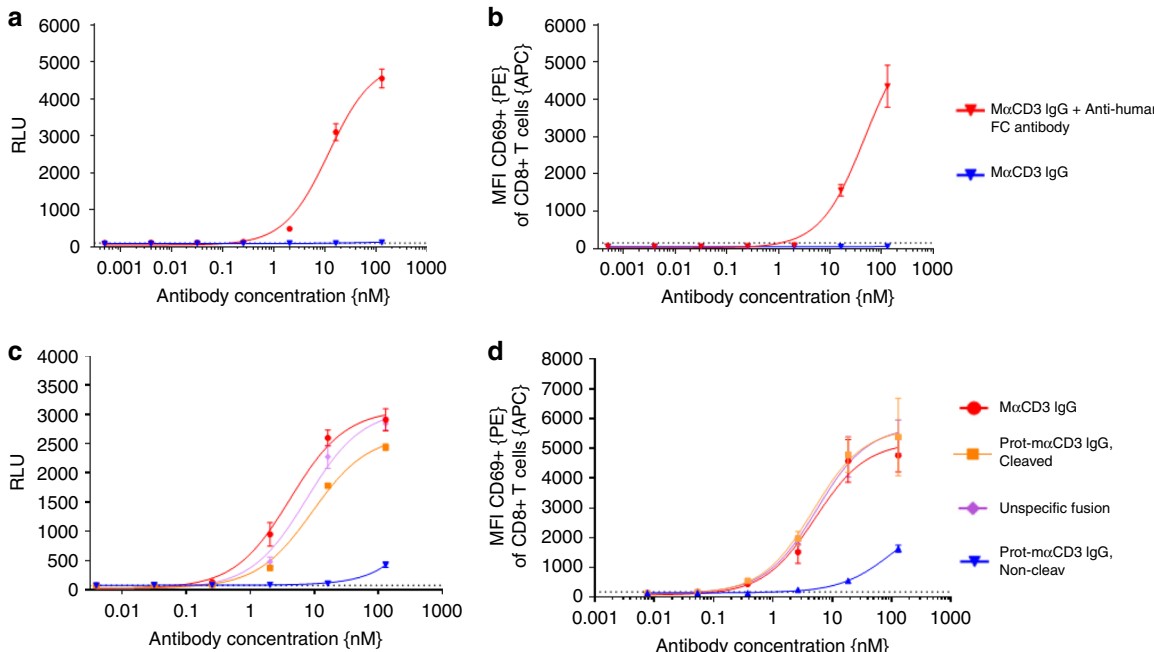

**Fig. 4 T-cell activation is reversibly impaired by blocking anti-CD3 binder.** T-cell/Jurkat NFAT activation is dependent on crosslinking of mα-CD3. **a** Monovalent anti-CD3 IgG is able to activate luciferase in a Jurkat NFAT luciferase reporter cell line after crosslinking by plate-coated anti-human Fc (red curve) whereas in the absence of crosslinking no luciferase activity can be detected (blue curve). The dotted line indicates the luminescence for Jurkat NFAT-cells without mα-CD3 IgG on plate-coated anti-human Fc. Each value represents the mean of triplicates, standard deviation is indicated by error bars (representative experiment of $n = 3$). **b** Monovalent α-CD3 IgG is able to activate CD8-positive T cells, measured by quantification of the early activation marker CD69, after crosslinking by plate-coated anti-human Fc (red curve) whereas in the absence of crosslinking no CD8-positive T-cell activation can be detected (blue curve). The median fluorescence intensity (MFI) for CD69 of CD8 T cells is shown. Each value represents the mean of triplicates, standard deviation is indicated by error bars (representative experiment of $n = 3$). **c, d** mαCD3 IgGs were bound to plate-coated anti-human Fc antibody before Jurkat NFAT reporter cells or PBMCs were added. Jurkat NFAT activation is measured in relative luminescence units (RLU) and T-cell activation was assessed by quantification of CD69 by FACS analysis. Cleavage of the Prot-mαCD3 IgG containing a matriptase cleavable linker was performed by incubation of Prot-mαCD3 IgG with purified recombinant human matriptase for 24 h at 37 °C. **c** Jurkat NFAT activation mediated by Prot-mαCD3 IgG. Cleaved Prot-mαCD3 IgG, blocked Prot-mαCD3 IgG, mαCD3 IgG and mαCD3 IgG with an N-terminal fusion of a nonspecific fusion (anti-CEA Fab) are shown. The dotted line indicates the luminescence for Jurkat NFAT-cells without any CD3 IgG. Each value represents the mean value of triplicates, standard deviation is indicated by error bars (representative experiment of $n = 3$). **d** The median fluorescence intensity (MFI) for CD69 of CD8 T cells is shown. Each value represents the mean value of triplicates, standard deviation is indicated by error bars (representative experiment for three different human PBMC donors).

antigen, we set up a method to analyze tumor samples without digestion to exclude artefacts coming from tumor digestion. This might be of importance, as not only tumor cells but also cells from the tumor microenvironment (e.g. tumor-associated macrophages (TAM)) are described to express MMP-2 and MMP-9[32]. Protease-activated TCBs containing linkers with different protease sites were incubated with mechanically cut tumor pieces before CD3-mediated T-cell activation was analyzed. In this setting, binding of the cleaved TCB to tumor cells and Jurkat NFA cells results in induction of a luciferase signal. Jurkat NFAT activation was detected for a benign FOLR1-positive ovary sample using the FOLR1-TCB but no activation was detected using the Prot-FOLR1-TCB (Fig. 7a). However Jurkat NFAT reporter cells were activated after incubation of FOLR1-positive ovarian tumor samples with FOLR1-TCB or Prot-FOLR1-TCB containing an MMP-matA site (Fig. 7b, c).

**Prot-FOLR1-TCB is efficacious in vivo.** As stability of Prot-FOLR1-TCB in human serum over time was shown (Supplementary Fig. 4), the stability of Prot-FOLR1-TCB was analyzed in vivo. Bioavailability of active Prot-FOLR1-TCB 7 days after intravenous injection revealed that the Prot-FOLR1-TCB containing the combined MMP-matA site was cleaved to some

extent (bioavailability around 35%) in non-tumor-bearing mice. Furthermore, we determined the serum bioavailability of the Prot-FOLR1-TCBs containing the single MMP site or the single matA site. Both were also cleaved to some extent (MMP ~25%, matA ~14%), however, less than the combined linker. In this analysis we included also a Prot-FOLR1-TCB with a new matriptase (matB) cleavage site and a matC site, described to be cleaved by matriptase[33] (Supplementary Table 2). Both Prot-FOLR1-TCBs had a very low serum bioavailability of ~5% in non-tumor-bearing mice (Fig. 8a). Thus, we used these Prot-FOLR1-TCBs in a subsequent efficacy study using a breast PDX (patient-derived xenograft) "BC004" model. Immunohistochemistry staining confirmed FOLR1 and matriptase expression in this PDX BC004 model (Fig. 8b). Tumor growth inhibition (TGI) was evaluated in an efficacy study in stem cell humanized NSG mice with autologous T cells and the respective TCBs at a dose of 4 mg/kg. FOLR1-TCB, Prot-FOLR1-TCB containing the matB site and the matC site showed significant TGI at day 62 compared to the vehicle group (Fig. 9a). The Prot-FOLR1-TCB containing a noncleavable (non-cleav) site was not different from vehicle showing that masking prevented antitumor efficacy. Comparing Prot-FOLR1-TCB with the matB site and the matC site, the latter one was more efficacious and comparable to the parental FOLR1-TCB (Fig. 9a).

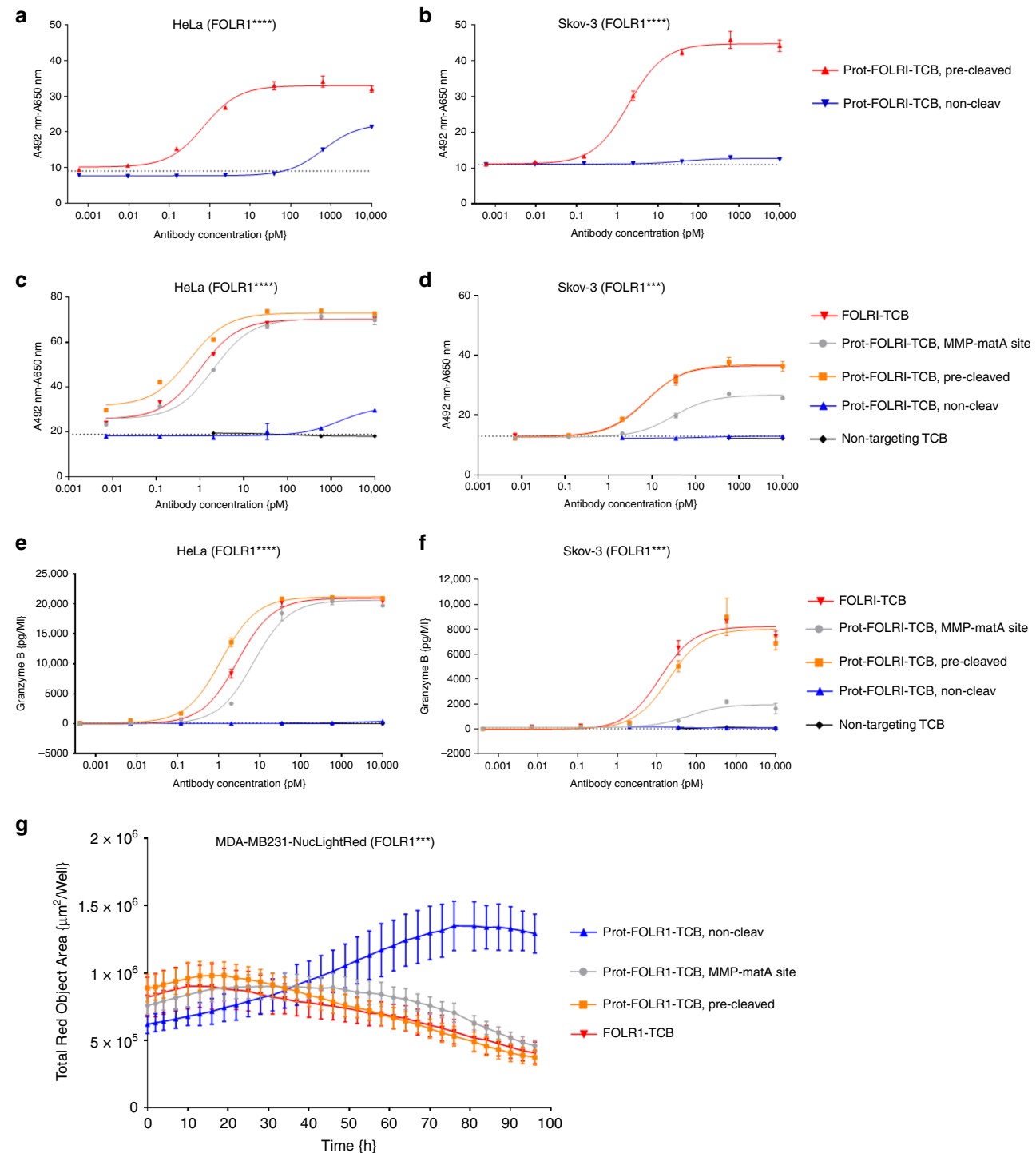

Immuno-pharmacodynamics was analyzed by quantification of human CD3 T cells in the tumor. All treatment groups were significantly different from the vehicle group (Fig. 9b), whereas the FOLR1-TCB and the Prot-FOLR1-TCB containing matB site were not significantly different. Prot-FOLR1-TCB containing matB site was also significantly different from Prot-FOLR1-TCB with noncleavable site, whereas the Prot-FOLR1-TCBs with matC site was not different from Prot-FOLR1-TCB with noncleavable site (Fig. 9b). Comparison of serum bioavailability of active Prot-FOLR1-TCB in non-tumor-bearing humanized mice and in tumor-bearing mice showed no evidence for tumor-leakage of activated Prot-FOLR1-TCB into the serum as

bioavailability of active Prot-FOLR1-TCB was comparable in non-tumor- vs. tumor-bearing mice (Fig. 9c, d).

## Discussion

CD3 targeting antibodies can be used for different purposes: the muromonab-CD3 antibody was approved to prevent allograft rejection after organ transplantation[34,35]. However, systemic T-cell activation led to side effects with strong cytokine release. Humanization and prevention of FcR binding improved the safety profile of anti-CD3 antibodies for autoimmune diseases but systemic T-cell activation remains a challenge. T-cell bispecific

**Fig. 5 Prot-FOLR1-TCB is efficiently blocked while its potency can be fully restored upon linker cleavage.** Dose–response curves for T-cell killing of FOLR1-positive tumor cells after 48 h mediated by TCB using PBMCs as effector cells with an E:T of 10:1. **a, b** Comparison of Prot-FOLR1-TCB (noncleavable and precleaved with matA site) for tumor cells with different FOLR1 expression levels (**a** HeLa cells, **b** Skov-3 cells). The cytotoxicity induced by the antibodies is shown. Each point represents the mean of triplicates. Standard deviation is indicated by error bars (representative experiment for one PBMC donor, $n = 2$ different human PBMC donors). **c, d** The cytotoxicity induced by the Prot-FOLR1-TCB containing MMP-matA site is shown. The Prot-FOLR1-TCB containing an MMP-matA site (gray circles) or a noncleavable linker (blue triangles pointing up) is shown. FOLR1-TCB (red triangles, pointing down), precleaved Prot-FOLR1-TCB (orange squares) and a nontargeted TCB (black circles) are used as controls. The cytotoxicity of the Prot-FOLR1-TCB with a noncleavable linker was only measured for four concentrations using Skov-3 cells as the target cells. The nontargeted TCB was only used at the three highest concentrations for both cell lines. The pretreatment of the Prot-FOLR1-TCB (orange squares) was done by incubation of Prot-FOLR1-TCB with recombinant human matriptase for 24 h at 37 °C. Each point represents the mean of triplicates. Standard deviation is indicated by error bars (representative experiment for one PBMC donor, $n = 3$ different human PBMC donors). **e, f** Granzyme B release was quantified by FACS analysis after incubation of PBMCs with 10 nM of Prot-FOLR1-TCB (cleavable linker vs. noncleavable linker) and FOLR1-positive target cells for 48 h. Each point represents the mean value of triplicates. Standard error is indicated by error bars (representative experiment for one PBMC donor, $n = 3$ different human PBMC donors). **g** FOLR1-positive tumor cell (CellPlayer™ MDA-MB-231 NucLight Red) growth inhibition mediated by Prot-FOLR1-TCB containing MMP-matA site and human PBMCs. Each point represents the mean of triplicates. Standard deviation is indicated by error bars (representative experiment for one PBMC donor, $n = 2$ different human PBMC donors).

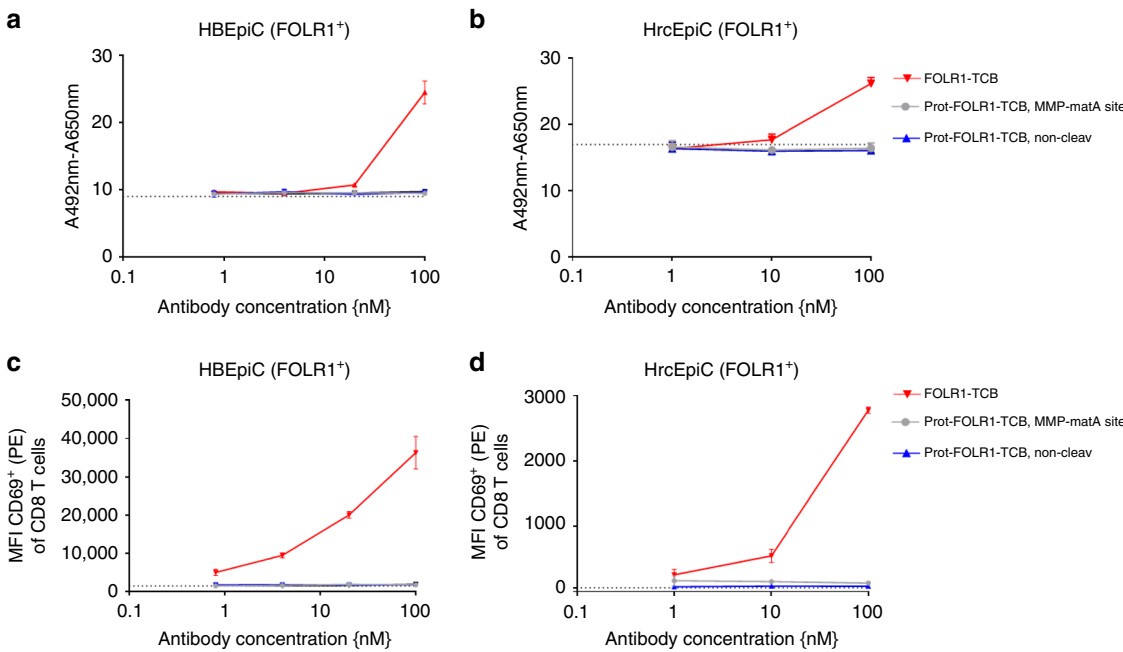

**Fig. 6 Cell cytotoxicity of FOLR1-TCB is abolished by blocking of anti-CD3 for primary cells with low FOLR1 expression.** T-cell killing of primary cells, expressing low levels of FOLR1, mediated by Prot-FOLR1-TCB with MMP-matA site using PBMCs as effector cells and an E:T ratio of 10:1. **a, b** Human bronchial epithelial cell (HBEpiC) (**a**) and human renal cortical epithelial cell (**b**) killing assessed after 72 h. Each point represents the mean of triplicates (one human PBMC donor shown, $n = 3$ different human PBMC donors). Standard deviation is indicated by error bars. **c, d** Median fluorescence intensity of CD69 of CD8-positive T cells is shown after incubation with TCB and HBEpiC (**c**) and HrcEpiC (**d**) cells. The dotted line indicates the MFI for CD8-positive T cells incubated without any TCB. Each point represents the mean value of triplicates, standard deviation is indicated by error bars (one human PBMC donor shown, $n = 3$ different human PBMC donors).

antibodies are promising agents to mediate potent tumor cell killing but they require a tumor antigen with expression restricted to tumor cells to avoid on-target/off-tumor toxicity on normal cells[24]. The development of TCBs for cancer immunotherapy may therefore profit from a protease-activated anti-CD3 moiety to reduce systemic side effects.

Here we describe a novel protease-activated $2 + 1$ Prot-FOLR1-TCB with a masked anti-CD3 Fab fragment and an inert Fc-region. T-cell activation requires binding of FOLR1-TCB to both FOLR1 and CD3. To avoid this in the periphery, the anti-CD3 Fab is blocked until activation by linker cleavage through a tumor-specific protease site (matriptase, MMP-2 or MMP-9) that connects the mask and anti-CD3 Fab. FOLR1 is overexpressed in ovarian[23,36] or lung cancer[23] but also expressed in normal tissue (like lung and kidney[23]) making it a suitable target for proof-of-

concept. FOLR1 targeting antibodies e.g. farletuzumab[37] and antibody–drug conjugates e.g. IMGN853 appear to be safe in in clinical trials[37–39]. However, targeting FOLR1 with TCBs resulted in on-target/off-tumor toxicity as FOLR1-TCBs can induce killing of normal cells with few hundred FOLR1 receptors. In line with this, toxicity was observed in non-human primates after injection of 10 μg/kg FOLR1-TCB[24] and in patients with ovarian cancer treated with the first-generation bispecific antibody OC/TR F(ab′)2 targeting FOLR1 and CD3[40]. Nevertheless, FOLR1-TCBs may be advantageous compared to FOLR1 antibodies due to their higher antitumor potency and NK-cell-independent mechanism[41].

The concept of tumor-specific protease-activation has been described previously (reviewed in refs. [42–44]). Active proteases located in the tumor microenvironment lead to activation of

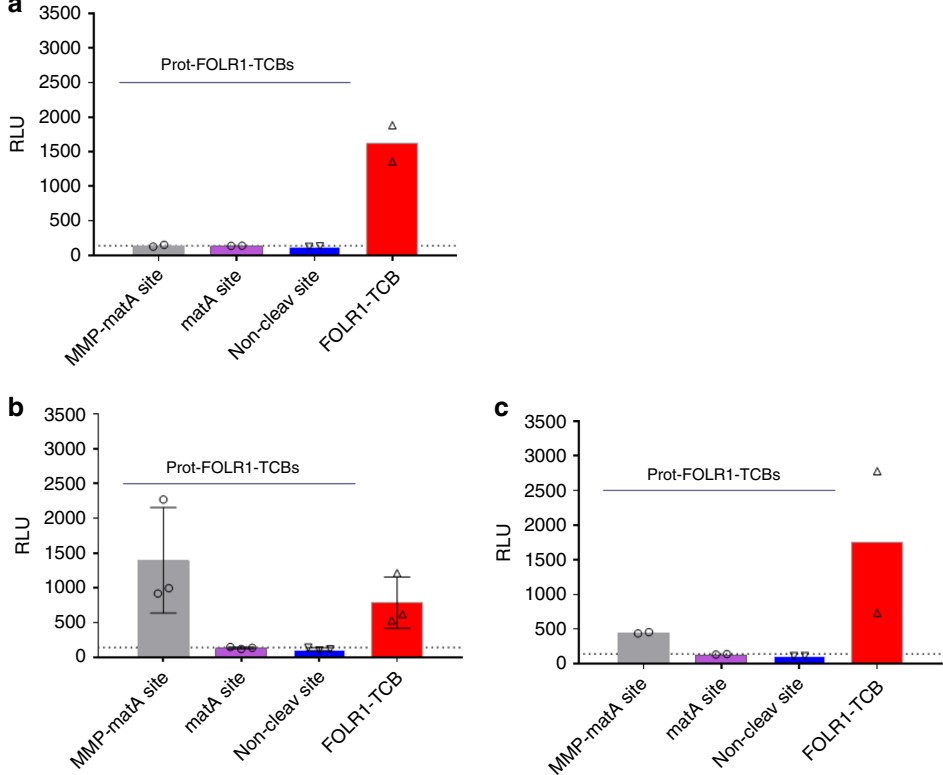

**Fig. 7 Prot-FOLR1-TCB can be activated by patient-derived ovary cancer explants.** Jurkat NFAT reporter assay was used to analyze activation of Prot-FOLR1-TCB (matA or MMP-matA linker) ex vivo by undigested human tumor explants. **a** Benign tumor of the ovary. **b** Cancer of the ovary. **c** Cancer of the ovary. Explants were mechanically cut and then incubated with TCBs and analyzed for CD3 activation using Jurkat NFAT cells. Jurkat NFAT activation is measured in relative luminescence units (RLU). Each symbol indicates the value measured for one biological sample incubated with Jurkat NFAT cells and Prot-FOLR1-TCB MMP-matA site (gray bar), matA site (purple bar), noncleavable site (blue bar) or FOLR1-TCB (red bar). **a** Each data point shows the mean of technical duplicates measured for one well ($n = 2$ biological replicates). **b** Each data point shows the value measured for one well ($n = 3$ biological replicates). Standard deviation is indicated by error bars. **c** Each data point shows the mean of technical duplicates measured for one well ($n = 2$ biological replicates). The dotted lines indicate luminescence for Jurkat NFAT-cells incubated with tumor samples but without any TCB.

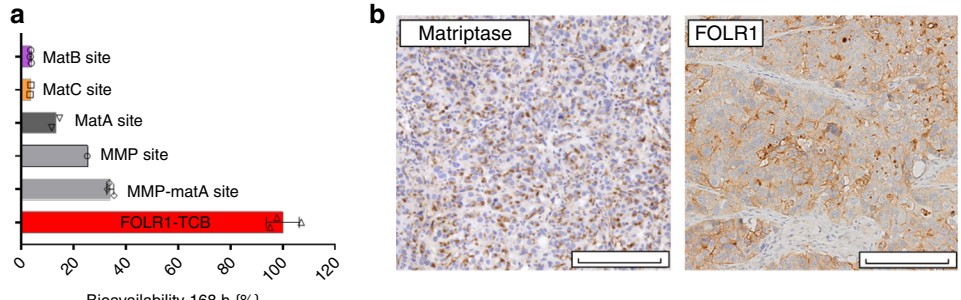

**Fig. 8 Stability of Prot-FOLR1-TCB depends on the cleavage site. a** Bioavailability of active Prot-FOLR1-TCB containing different cleavage sites at day 7 after intravenous single-dose injection in non-tumor-bearing NSG mice ($n = 3$ per group). Active and total Prot-FOLR1-TCB were quantified by ELISA using an anti-PG-LALA antibody (total Prot-FOLR1-TCB) and the anti-idiotypic anti-CD3 antibody (active Prot-FOLR1-TCB). Pharmacokinetic evaluation was conducted by noncompartmental methods. Areas under the serum concentration−time curve were calculated by linear trapezoidal rule. Bioavailabilities $F$ of the active FOLR1-TCB after Prot-FOLR1-TCB administration were calculated by comparing AUC 0−168 h values of FOLR1-TCB following i.v. administration of the respective pro-TCB (AUC from Prot-FOLR1-TCB) and administration of the active TCB (AUC FOLR1-TCB) according to $F(\%) = (AUC$ from Prot-TCB/AUC TCB) × 100. Dose corrections were not required, as equimolar doses of Prot-FOLR1-TCB and FOLR1-TCB were used in the respective studies. **b** Immunohistochemistry of FOLR1 and matriptase in breast PDX tumor at baseline. Some tumors were harvested at start of treatment for the baseline characterization of FOLR1 target antigen and matriptase by immunohistochemistry. Positive staining is observed as a brown precipitate within the sections. The evaluation of FOLR1 and Matriptase expression in the described in vivo model has been conducted twice in two independent samples. Scale bars indicates 200 μm and is included in the image.

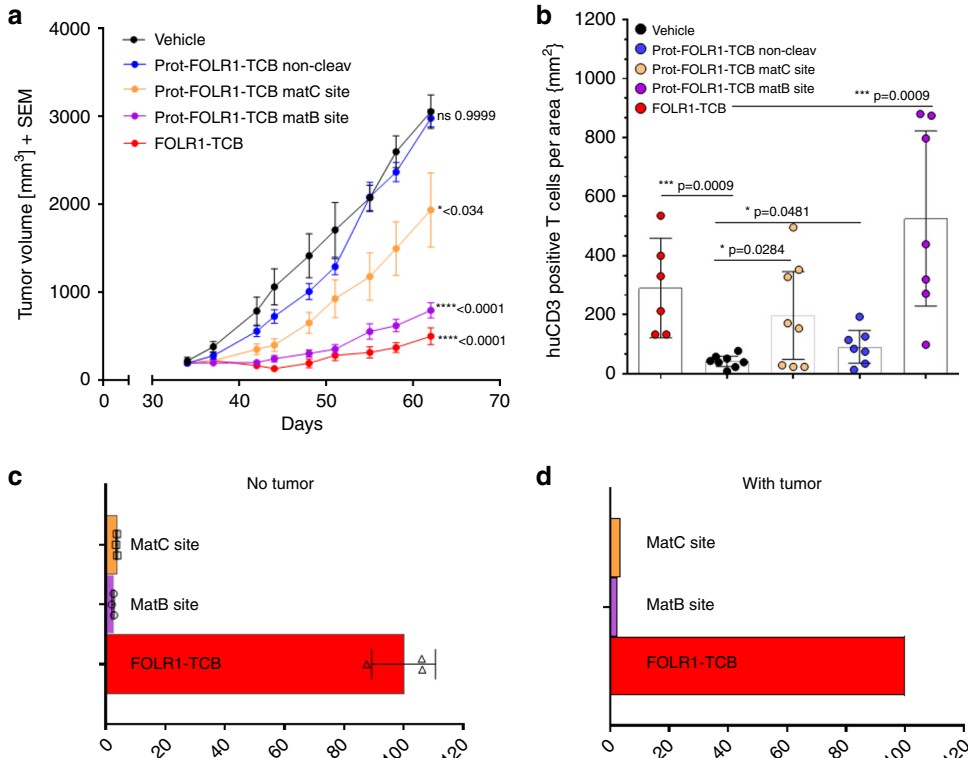

**Fig. 9 Prot-FOLR1-TCB is efficacious in vivo while there is no hint for tumor-leakage of activated Prot-FOLR1-TCB. a** Tumor growth inhibition curves of breast PDX BC004 model in humanized mice. Humanized mice ($n = 9$ per group) were weekly i.v. injected with equimolar doses of Prot-FOLR1-TCBs (4 mg/kg) containing different cleavage sites, FOLR1-TCB (3.6 mg/kg) or vehicle. Each dot represents the mean tumor volume ± SEM. Efficacy was evaluated by measuring the reduction of the mean tumor volume at day 62 relative to vehicle control. Statistical analysis was done using one-way ANOVA with Tukey−Kramer correction. No significant efficacy was observed for the Prot-FOLR1-TCB with the noncleavable linker comparing to vehicle control. Significant tumor growth inhibition was induced by the FOLR1-TCB (****$p < 0.0001$), the Prot-FOLR1-TCB with the matB site (****$p < 0.0001$) and the Prot-FOLR1-TCB with matC site (*$p = 0.034$) all compared to vehicle group. The in vivo efficacy study has been conducted once ($n = 9$ mice per group; $n = 8$ mice for matB group). **b** Quantification of human CD3-positive T cells for the different treatment groups of efficacy study using breast PDX in humanized mice. All data points shown in bar chart. Each data point represents the value for one mouse. 95% confidence interval is shown for each group. Two-tailed, unpaired $t$ test was used to calculate statistics. Significantly more huCD3 T cells per mm³ were found for animals treated with FOLR1-TCB (***$p = 0.0009$) and Prot-FOLR1-TCBs (matC *$p = 0.0284$ and matB site ***$p = 0.0009$) than for vehicle group. However also the Prot-FOLR1-TCB with a noncleavable linker had significant more huCD3 T cells per area compared to vehicle (*$p = 0.0481$). Representative images are shown in Supplementary Fig. 9. **c, d** Bioavailability of active Prot-FOLR1-TCBs with different linkers in non-tumor-bearing (**c**) and tumor-bearing humanized mice (**d**). Bioavailability of active Prot-FOLR1-TCB containing different cleavage sites at day 7 after injection in non-tumor-bearing humanized NSG mice ($n = 3$ per group) or breast PDX tumor-bearing humanized mice ($n = 6$ per group). Bioavailabilities F of the active FOLR1-TCB after Prot-FOLR1-TCB administration were calculated (as described in Fig. 8) by comparing AUC 0−168 h values of FOLR1-TCB following i.v. administration of the respective pro-TCB (AUC from Prot-FOLR1-TCB) and administration of the active TCB (AUC FOLR1-TCB) according to $F(\%) = (\text{AUC from Prot-TCB}/\text{AUC TCB}) \times 100$. For tumor-bearing mice AUC were calculated from composite concentration−time data ($n = 3$/time point). Dose corrections were not required, as equimolar doses of Prot-FOLR1-TCB and FOLR1-TCB were used in the respective studies.

tumor targeting moieties by cleavage of the linker and consecutive unmasking of the targeting moiety.

Proof-of-concept was shown for tumor-specific protease-activation of a prodrug that can reduce on-target toxicity for an EGFR-targeted antibody with N-terminally fused blocking peptides and a protease cleavable linker[33,45]. Other examples are an anti-PD-L1 probody CD-71 targeting probody drug conjugate[46,47]. Similarly, this approach was described for a protease-activated CTLA-4 antibody[48] and using a coiled-coil masking domain for CD20, HER2 and CD3 antibodies[49] This strategy of protease-activation demonstrated enhanced tolerability in (pre-) clinical studies while retaining antitumoral efficacy.

Here, we chose not to block the tumor targeting moiety, but rather the anti-CD3 targeting moiety in a TCB in order to generate a platform applicable to various tumor antigens. Contrary to peptide blocking moieties the anti-idiotypic anti-CD3 scFvs, described here, can be humanized decreasing the risk of immunogenicity[50,51].

Our results show that it is feasible to block CD3 binding of anti-CD3 Fab fragments in IgG-like antibodies by an anti-idiotypic disulfide-stabilized anti-CD3 scFv and to unblock this by tumor-specific proteases[52].

First, we generated a monovalent anti-CD3 IgG with one anti-idiotypic anti-CD3 scFv with an affinity of approx. 2 nM. When the scFv was attached to both anti-CD3 Fab fragments in a bivalent homodimeric IgG (Supplementary Fig. 7), the corresponding construct could not be properly purified due to its high aggregation tendency following pH neutralization after affinity chromatography, which is likely a consequence of intramolecular aggregation of scFvs. The globular structure of Prot-mαCD3 IgG and the validity of the matriptase-triggered de-masking concept is substantiated by single particle analytics. NS-TEM data of complexes made from CD3 antigen and activated Prot-mαCD3 IgG confirm that the distal region is crucial for complex formation. Matriptase-driven unmasking

demonstrated individual molecule level, in situ, with tapping-mode AFM.

In a second step blocking of the anti-CD3 moiety by the anti-idiotypic, anti-CD3 scFv could be also shown in the TCB format. Target cell cytotoxicity was significantly reduced for the Prot-FOLR1-TCB containing a noncleavable site. Notably, the masking efficiency, in the Prot-TCB format correlated with the antigen expression level of the target cells and with the affinity of the anti-idiotypic anti-CD3 binder to the anti-CD3 in the TCB.

Different proteases are described to be active in several cancers like serine and cysteine proteases as well as matrix metalloproteinases[8,53–55]. Typically, the expression and the activity of these proteases is minimal in normal tissue, so that these proteases can be exploited for tumor-specific activation or imaging. For proof-of-concept we have focused on MMP-2, -9 and matriptase for the Prot-FOLR1-TCB as these proteases are known to be overexpressed and active in ovarian carcinoma[9,10,12,56] with minimal activity in normal tissue[20,33,53]. LeBeau et al.[21] showed that matriptase is expressed in normal colon, but the active form of matriptase was not detected there. Demeter et al.[20] showed that both MMP-2 and MMP-9 are not active in the serum of patients but are active in the ascites and tumors of recurrent patients with EOC. To prove the stability of our protease-activated antibodies, we showed the integrity after incubation in human serum for 14 days at 37 °C. Two different matriptase cleavage sites were introduced into the generated constructs. For the construct containing the RQRRVVGG matriptase cleavage site, we observed that it was cleaved during production in HEK293 cells which was attributed to a furin cleavage site within this linker.

Comparing Prot-FOLR1-TCB using different linkers solely cleaved by cellular proteases, we confirmed a synergistic effect for the combination of the cleavage sites for MMP-2, -9-matriptase (MMP-matA) compared to matriptase (matA) or MMP-2, -9 (MMP) linkers alone. Several linkers described for protease-activated antibodies currently under investigation also contain 2 −3 substrate sequences for different proteases. One example is an FAP-CD95L fusion protein containing an MMP-2/uPA cleavable linker[57]. Notably, the potency of the Prot-FOLR1-TCB containing an MMP-2, -9-matriptase linker was comparable to FOLR1-TCB using HeLa or Skov-3 cells as target cells. Target cell lysis correlated with T-cell activation and secretion of granzyme B induced by the crosslinking through FOLR1-positive cancer cells as it was recently shown for CEA TCB2.

For primary human bronchial epithelial cells and primary human renal cortical cells expressing low amounts of FOLR1, the masking significantly reduced target cell killing even at concentrations tenfold higher than the one used for tumor cells. Reduced on-target toxicity was also shown for the EGFR-targeting Probody described by Desnoyers et al.[33] in non-human primates compared to cetuximab. Similarly, Watermann et al. observed reduced liver-toxicity mediated by the protease-activated FAP-CD95L fusion protein compared to the parental molecule. These data provide evidence that cleavable masks can enhance safety of compounds targeting antigens whose expression is not restricted to the tumor cells.

Importantly, the potency of the Prot-FOLR1-TCB was fully recovered after cleavage with recombinant human matriptase. Additionally, we showed activity of the Prot-FOLR1-TCB containing MMP-2, -9-matriptase cleavable linker in undigested human ovarian tumor samples. For this purpose we applied undigested tumor samples as it was reported that cells of the tumor microenvironment are involved in protease expression e.g. TAM or fibroblasts[58–61]. In order to check for FOLR1 and protease expression in this assay without the need to monitor pre-existing T cells and interference from debris and dead cells common in human tumor explants, we developed an assay based

on Jurkat cells expressing luciferase under the control of an NFAT-inducible reporter. Activation of the Prot-FOLR1-TCB containing an MMP-matA site was observed for two undigested FOLR1 + ovarian carcinoma samples whereas no activation occured in a benign FOLR1 + ovarian sample. This is in line with reports describing higher MMP-2 and MMP-9 activity in advanced EOC or metastasis compared to benign tumors[18,20] and their role in extracellular matrix degradation and activation of growth factors to facilitate invasion and tumor growth[32,55,62]. To demonstrate the applicability of the protease-activated anti-CD3 Fab fragment for other targets, we engineered an analoguous protease-activated mesothelin TCB (Prot-MSLN-TCB). We confirmed efficient masking of the Prot-MSLN-TCB with a non-cleavable linker and comparable efficacy of the activated Prot-MSLN-TCB and the MSLN-TCB using MSLN-positive target cells.

Finally, aiming to analyze the Prot-FOLR1-TCB containing MMP-matA site in vivo, we first tested the stability in non-tumor-bearing mice. We observed that the Prot-FOLR1-TCBs with MMP, matA and MMP-matA sites were cleaved to some extent in the absence of tumor in vivo. The Prot-FOLR1-TCBs containing the single cleavage sites were cleaved to a lower extent (MMP ~25%, matA ~14%) while the combination of both cleavage sites resulted in enhanced cleavage (combined MMP-matA ~35%). This finding was unexpected as all linkers were stable in human serum. In order to find a suitable cleavage site, we analyzed the in vivo stability of Prot-FOLR1-TCB containing the matriptase site matB (PMAKK) or the matriptase site matC (LSGRSDNH) which was described to be stable in cynomolgous monkeys as a positive control[33]. Both Prot-FOLR1-TCBs containing matB or the matC site had low serum bioavailability of ~5% in non-tumor-bearing mice and thus were evaluated for their antitumor activity in vivo. Tumor growth inhibition was evaluated in an efficacy study using an orthotopic breast PDX BC004 model in humanized mice with autologous T cells. Prot-FOLR1-TCB containing a noncleavable linker behaved comparable to vehicle regarding TGI whereas the FOLR1-TCB, Prot-FOLR1-TCB containing the matB site and the matC site were significantly different from vehicle. The Prot-FOLR1-TCB containing the matB site induced superior TGI from all Prot-FOLR1-TCBs, suggesting that the matriptase cleavage of this site may be more efficient than the matriptase cleavage of matC. Importantly, the comparison of bioavailabilities of active Prot-FOLR1-TCB in non-tumor-bearing and tumor-bearing mice did not suggest any tumor leakage of Prot-FOLR1-TCB from the tumor into the serum. As an additional safety measure for targets that are expressed to a low extent on normal cells, this novel protease-activated anti-CD3 moiety could be used to engineer TCBs for improved specificity and therefore increase the number of targets amenable for TCBs.

## Methods

**Construction of protease-activated antibodies.** The variable chains of the scFvs were connected by a $(G_4S)_4$ linker and cysteins were inserted (VH44-VL100) for disulfide stabilization[25,26]. Single chain variable fragment (scFv) sequence synthesis was ordered at Invitrogen, including the necessary restriction sites for cloning. Single chain Fv DNA sequences of three anti-idiotypic anti-CD3 antibodies were N-terminally fused in frame with the anti-CD3 Fab-Fc chain preinserted into the respective recipient mammalian expression vector. The construction of expression vectors for TCBs was performed according to standard recombinant DNA technologies. All antibody chain genes were separately inserted into expression vectors under the control of the MPSV or CMV promoter (myeloproliferative sarcoma virus or cytomegalovirus) and transiently expressed in HEK293 cells. The anti-idiotypic single chain fragments (scFv) were fused to the anti-CD3 variable heavy chain (VH) in the respective chain of TCB. In order to get high yields of correctly paired molecules, the "knobs-into-holes" (KiH) technology was used for hetero-dimerization[28]. P329G, L234A and L235A (PG LALA) mutations were inserted in CH3 and CH2 to prevent binding to FcγRs and C1q[63]. For cases when no common light chain could be used (MSLN-TCB and Prot-MSLN-TCB), a CrossMAb[VH-VL]

format[29,32] and charged residues in constant chains[31] were used to assure correct light chain pairing.

**Cell lines.** HEK293, HeLa and Skov-3 cells were purchased from ATCC (American Type Culture Collection (ATCC)), MDA-MB-231 NLR were purchased from Essen Bioscience (Cat.# 4487), human bronchial epithelial cells (HBEpiC, 3210) and human renal cortical epithelial cells (HrcEpiC, 4110) were purchased from ScienCell Research Laboratories, AsPC-1 (ECACC, 96020930) cells were obtained from the European Collection of Cell Cultures (ECACC) and NCI H596 cells were provided by Roche Innovation Center Munich. Jurkat NFAT-cells were purchased from Promega. All cells were routinely cultured at 37 °C and 5% $CO_2$ and tested for mycoplasma contamination. The cell identity of all tumor cell lines was verified by FTA cell authentication service provided by the ATCC[64]. Antigen-binding sites were determined using QIFIKIT® (Dako) according to the manufacturer's instructions.

**Killing assays.** Human peripheral blood mononuclear cells (PBMCs) were purified from buffy coats of healthy donors, obtained from Blutspende Zürich SRK, by conventional histopaque gradient (Sigma-Aldrich). Blutspende Zürich SRK confirmed that all donors consented into the use of the sample for research purpose. Adherent target cells were trypsinized (0.05 % trypsin/EDTA; Gibco) and counted using a Vi-CELL device (Beckman Coulter). 20,000 target cells per well were seeded in flat-bottom 96-well plates (tissue culture test plates from TPP) and incubated for approx. 20 h at 37 °C, 5% $CO_2$ before antibodies and human PBMC effector cells were added (E:T ratio of 10:1). Target cell killing was measured after 48/72 h of incubation at 37 °C, 5% $CO_2$ by quantification of lactate dehydrogenase (LDH) release into cell supernatants by dead cells (LDH detection kit; Roche Applied Science). Minimal lysis refers to target cells incubated with effector cells without any TCB. T-cell activation was analyzed by quantification of CD69 for CD8-positive T cells using a MACSQuant device (Miltenyi Biotec)[65]. Cytokine (IFN-γ, TNF-α, IL-2, GM-CSF) and cytotoxic granule (granzyme B) secretion was assessed 48 h after incubation of target cells with TCB and PBMCs (as above) using Human Soluble Protein Master Buffer Kit (BD Biosciences) according to the manufacturer's protocol. The cleavage of the Prot-FOLR1-TCB was done by incubation of 1 μl of purified recombinant human matriptase (0.44 mg/ml, R&D Systems) with approx. 10 nmol of antibody in histidine buffer (20 mmol/l, pH 6, Bichsel) for 24 h at 37 °C. The precleaved TCB was not purified after incubation.

**Kinetic of tumor cell growth using IncuCyte.** Tumor cell growth inhibition mediated by Prot-FOLR1-TCB was assessed on CellPlayer™ MDA-MB-231 Nuc-Light Red cells (Essen BioScience) naturally expressing FOLR1. Adherent target cells were trypsinized (0.05% trypsin/EDTA; Gibco) and 5000 cells per well were seeded in flat-bottom 96-well plates (tissue culture test plates from TPP) before molecules and human PBMC effector cells were added (E:T ratio of 10:1). All samples were performed in triplicates and incubated in IncuCyte (Essen BioScience) at 37 °C, 5% $CO_2$. Tumor cell growth was analyzed by target-cell count. The first scan that was carried out approx. 2 h after the addition of PBMCs and TCBs is indicated as time 0 h. The cleavage of the Prot-FOLR1-TCB was done by incubation of 1 μl of purified recombinant human matriptase (0.44 mg/ml, R&D Systems) with approx. 10 nmol of antibody in histidine buffer (20 mmol/l, pH 6, Bichsel) for 24 h at 37 °C. The precleaved TCB was not purified after incubation.

**Reporter assay using patient-derived cancer explants.** We used a Jurkat-NFAT reporter cell line (Promega) to check target expression (FOLR1) and protease activity in patient-derived undigested human tumor samples. Tumor samples (Indivumed GmbH, Germany) were shipped overnight in transport medium. Approximately, 24 h after surgery, the sample was cut into small pieces (<1 mm in diameter) before 2−3 pieces were placed into wells before 50 nM of TCBs was added. In one experiment (Fig. 7a), the pieces were put in 24-well plates prepared with Millicell Cell Culture Insert, 12 mm, hydrophilic PTFE, 0.4 μm (PICM01250, MerckMillipore); in the other experiment (Fig. 7b, c) the 2−3 pieces were put in wells of a 96-well plate prepared with matrigel (Corning/VWR). Pieces were covered with Matrigel and hardened for 2 min at 37 °C. 50 nM of TCBs was incubated with tumor pieces for 48 h at 37 °C, 5% $CO_2$. Jurkat-NFAT reporter cells were harvested and viability was assessed using ViCell. 500,000 Jurkat NFAT-cells/well were added for 24-well plate and 50,000 Jurkat NFAT-cells/well were added for 96-well plate. The plates were incubated for 5 h at 37 °C in a humidified incubator before ONE-Glo substrate solution (Promega) was added to each well and incubated for 10 min at room temperature in the dark. Luminescence was detected using WALLAC Victor3 ELISA reader (PerkinElmer2030), 1 s/well as detection time.

**Mouse model.** All mice were maintained under specific pathogen-free condition with daily cycles of 12 h light/12 h darkness. The animal facility has been accredited by the Association for Assessment and Accreditation of Laboratory Animal Care (AAALAC). All animal studies were performed in accordance with the Federation for Laboratory Animal Science Associations (FELASA). The animal studies were approved by and done under license from the Government of Upper Bavaria (Regierung von Oberbayern; Approval number: Az 55.2.1.54-2532.0-10-16).

Animals were maintained for 1 week after arrival to get accustomed to the new environment and for observation. Daily continuous health monitoring and weekly body weight measurement was conducted.

Hematopoietic stem cell humanized mice (humanized mice), used for efficacy or single-dose PK studies, were generated in-house. Briefly, 4–5-week-old female NOD scid gamma (NSG) mice (Jackson Laboratory, Sacramento, CA USA) were injected i.p. with 15 mg/kg Busulfan (Busilvex, Pierre Fabre Limited) in a total volume of 200 μl. Twenty-four hours later, mice were injected intravenously (i.v.) with $1 \times 10^5$ CD34+ cord blood cells (STEMCELL Technologies Inc, Grenoble, France). Fifteen weeks after cell injection, mice were bled and screened for successful humanization by flow cytometry.

In some single-dose PK studies, 8−10-week-old female NOD scid gamma (NSG) mice (Jackson Laboratory, Sacramento, CA USA) were used without any humanization.

**Single-dose PK and stability study.** NSG or non-tumor-bearing and tumor-bearing humanized NSG mice received a single injection of different antibodies (equimolar doses). Seven days post infusion mice were bled under anesthesia (retro-orbital). Fresh blood was collected in serum separator tubes (Sarstedt, Nuembrecht, Germany) and after centrifugation, serum was frozen and stored at −20 °C for further analysis.

**Patient-derived xenograft (PDX) model.** The human breast cancer patient-derived xenograft HER2+ ER− xenograft model BC_004 was purchased from OncoTest (Freiburg, Germany). Tumor fragments were digested with Collagenase D and DNase I (Roche), counted and $2 \times 10^6$ BC004 cells were injected in total volume of 20 μl PBS into the mammary fat pad. Treatment was started once tumors reached an average volume of approximately 200 mm³.

**Therapeutic antibody treatment.** A total of nine animals were assigned per group. No statistical methods were used to predetermine the total number of animals needed for this study; however, taking into consideration the heterogeneity of tumors growth as well the heterogeneous humanization rate of NSG mice, we experienced nine mice per group as a good number for statistical power. All mice were injected i.v. with 200 μl of the appropriate solution. The mice in the vehicle group were injected i.v. with Histidine buffer (20 mM Histidine, 140 mM NaCl, pH 6.0) and the treatment group with the antibody diluted with Histidine buffer to a volume of 200 μl. Mice received once weekly injections of 4 mg/kg of each compound (equimolar doses, so 3.6 mg/kg of TCB) and a total of four treatments.

**Tumor volume measurement.** Tumor volume (½ [length × width²]) was measured three times per week by caliper.

**Necropsy and immunohistochemistry.** At study termination, mice were sacrificed and tumors were surgically removed from all animals. Some tumors were harvested at start of treatment for the baseline characterization by immunohistochemistry. All tissue samples were fixed in 10% formalin (Sigma, Germany) and processed for FFPET (Leica 1020, Germany). Four-micrometer paraffin sections were subsequently cut in a microtome (Leica RM2235, Germany). Human Matriptase immunohistochemistry was performed with anti-human ST-14 (PA5-29764 from Thermo Scientific, Germany), human folate receptor alpha with anti-FOLR1 (BN3.2, Byosystems, Switzerland) and human T-cell detection with anti-CD3 (ab5690, Abcam, Germany). Stainings were performed in the Leica autostainer (Leica ST5010, Germany) following the manufacturer's protocols. Sections were counterstained with hematoxylin (Sigma-Aldrich) and slides were scanned using Olympus VS120-L100 Virtual Slide Microscope scanner. Quantification of human CD3-positive cells from scan images was performed with Definiens software (Definiens, Germany). For this, whole scans were uploaded in the tissue developer module and necrotic areas were excluded with segmentation analysis. Secondly, a threshold was set to recognize the brown staining of the targeted CD3 T cells and subsequently the algorithm for cell quantification was automatically run. The output data for CD3 quantification were then transferred to GraphPad Prism for analysis of significance.

**Bioanalytics of serum samples.** For PK assessments a specific ELISA was developed for specific analysis of CD3 binding competent drug (active TCB) in the presence of Prot-FOLR1-TCBs. Unbound proteins were washed away after each step of reagent or sample addition (stepwise assay protocol). An anti CD3-binding site-specific monoclonal antibody was coated onto a microtiter plate followed by sample addition. Bound analyte was then incubated with digoxigenin-labeled Fc-specific detection antibody (binding to a mutation on the Fc part of the analyte), followed by an incubation with anti-digoxigenin Fab fragment conjugated to horseradish peroxidase. Finally, formed immobilized immune complexes were visualized by addition of HPPA (3-(4-Hydroxyphenyl)propionic acid) solution, a fluorogenic POD substrate. The fluorescence intensity was measured with an excitation wavelength of 320 nm (25 nm bandwidth) and an emission wavelength 400 nm (20 nm bandwidth).

Pharmacokinetic evaluation was conducted by noncompartmental methods. Areas under the serum concentration-time curve (AUC) were calculated by linear trapezoidal rule. Bioavailabilities $F$ of the active TCB after Prot-FOLR1-TCB administration were calculated by comparing AUC $0-168$ h values of the active FOLR1-TCB following i.v. administration of the respective Prot-FOLR1-TCB (AUC from pro-TCB) and administration of the active TCB (AUC FOLR1-TCB) according to $F(\%) = (\text{AUC from pro-TCB/AUC TCB}) \times 100$. Dose corrections were not required, as equimolar doses of Prot-FOLR1-TCB and active FOLR1-TCB were used in the respective studies.

**Statistical analysis.** For statistical analysis of the in vivo TGI curves, one-way ANOVA Tukey−Kramer correction was used. CD3 T-cell counts were analyzed using two-tailed, unpaired $t$ test using GraphPad Prism 6 Software.

$p$ values below 0.05 were considered as significant and were indicated with asterisks (n.s. $p > 0.05$; $*p \leq 0.05$; $**p \leq 0.01$; $***p \leq 0.001$, $****p \leq 0.0001$). Exact $p$ values are shown in the figures.

**Reporting summary.** Further information on research design is available in the Nature Research Reporting Summary linked to this article.

## Data availability

All data generated or analyzed during this study are included in this published article (and its supplementary information files). Data underlying Figs. 4, 5, 6, 7, 8a, and 9, Supplementary Figs. 1, 3, and 6 are provided as Source Data file. Material might be obtained for research purposes only under Material transfer agreement (MTA).

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

## Acknowledgements

We thank Erwin van Puijenbroeck, Fabian Birzele, Alexander Bujotzek, Thomas O'Brien, Christa Bayer, Brian Steiner, Inja Waldhauer, Linda Fahrni, Christian Müller, Manuel Späni, Michaela Ketterer, Sara Colombetti and Marina Bacac for their help. S.K. and S.E. are supported by grants from the Wilhelm Sander Stiftung (grant number 2014.018.1 to S.E. and S.K.), the international doctoral program "i-Target: Immunotargeting of cancer" funded by the Elite Network of Bavaria (to S.K. and S.E.), the Melanoma Research Alliance (grant number N269626 to S.E. and 409510 to S.K.), the Marie-Sklodowska-Curie "Training Network for the Immunotherapy of Cancer (IMMUTRAIN)" funded by the H2020 program of the European Union (to S.E. and S.K.), the Else Kröner-Fresenius-Stiftung (to S.K.), the German Cancer Aid (to S.K.), the Ernst-Jung-Stiftung (to S.K.), by LMU Munich's Institutional Strategy LMUexcellent within the framework of the German Excellence Initiative (to S.E. and S.K.), the Bundesministerium für Bildung und Forschung (project Oncoattract to S.E. and S.K.), the Deutsche Forschungsgemeinschaft, the José-Carreras Leukämie Stiftung, the Hector-Foundation (all to S.K.) and the European Research Council (ERC, grant 756017, ARMOR-T to S.K.).

## Author contributions

Conception and design: M.G., P.U., C.K. and P.B. Development of methodology and data acquisition: M.G., W.F.R., G.J.; S.G.-R., A.F.-G., J.P., E.S., M.E.L., K.-G.S., M.R., V.N., H.S. and P.R. Analysis and interpretation of data: M.G., C.K., P.B., W.F.R., J.S., V.N., C.H., J.E., A.F.-G., M.E.L., M.R., H.S. and P.R. Writing and review of the manuscript: M.G., C.K., P.B., J.S., W.F.R., A.F.-G., S.K. Administrative, technical, or material support: M.G., K.-G.S., M.R. Study supervision: C.K., P.B., S.E., S.K., P.U.

## Competing interests

The authors declare the following competing interests. Parts of this work have been performed for the doctoral thesis of M.G. associated to the international doctoral program "i-Target" at the Ludwig-Maximilians-Universität München. M.G., K.-G.S., A.F.-G., M.R., M.E.L., J.S., J.E., C.H., W.F.R., G.J., V.N., P.U., P.B. and C.K. are employees of Roche. J.P. and E.S. are employees and hold ownership in Nimble Therapeutics. J.P. and E.S. own Nimble Therapeutics stock. C.K., P.B., A.F.-G., P.U., K.-G.S., M.G., E.S., J.P. are inventors in patent applications related to this work. C.K., P.U., M.G., P.B., W.F.R., M.R., G.J., S.G.R., J.S., A.F.-G. own Roche stock.
