## [Peer Review File · Nature Communications]

Reviewers' Comments:

Reviewer #1:

Remarks to the Author:

The authors describe a method of masking the anti CD3-specificity of a T cell bispecific antibody (TCB) using an anti-idiotypic anti-CD3 domain that interferes with the normal anti-CD3 binding function of the TCB. This technique enabled the authors to "activate" the TCB by using a cleavable linker between the two domains that is cleaved by proteases naturally expressed by tumor cells that express FOLR1. Although the technique of masking antibodies has been previously described (e.g. Tzou et al 2017), this manuscript is the first to describe the technique in detail using a TCB. I recommend this paper be accepted pending moderate revisions. My main concerns are:

1. Grammar and sentence flow should be improved. Especially paragraphs in the discussion seem disjointed.
2. Some figure titles describe the outcome and some merely mention the assay used. Amend titles to convey main message
3. HrcEpiC cells are not listed in the materials section
4. For most figures, a consistent dose response curve is shown. In figure 7 and 8, only single antibody doses are used. Please explain the rationale for this. A dose response curve provides significantly more information and would benefit readers.
5. At the end of the conclusion, the sentence about MABEL seems out of place.
6. On page 8, the last two paragraphs start with "Table 1)", which did not make sense to me.
7. Figure 3B: It is unclear what kind of statistics was done here and whether the provided images are simple representative images.
8. In figure 4C-D: It would help the reader to use the word "unspecific fusion" in the figure description.
9. Throughout the manuscript, different Prot-FOLR1-TCBs are used (with different linkers). Especially in figure 5A, but also in the other figures and in the text throughout the manuscript, it is unclear which linker is used.

Reviewer #2:

Remarks to the Author:

In this manuscript the authors describe an interesting T cell Targeting bi-specific format (Prot-TCB) for cancer immune-therapy. The concept of attaching an anti-idiotypic anti-CD3 scFV with a protease sensitive linkers to mask the anti-CD3 binding domain is intriguing. The authors provide excellent and detailed information on arriving at the optimum format design and showing biophysical/biochemical, structural (NS-TEM) and in vitro functional characterization (CD3 domain activation by r-protease, cell lines and fresh tumor potentially expressing MMP 2, 9 and/or matriptase) for this new "pro-drug" format (Prot-TCB). They also provide data for in vitro serum stability and preliminary manufacturability for Prot-TCB format. The authors suggest that the main advantage of such a "pro-drug" format will be tumor site specific activation of such molecules that will potentially enhance safety and reduce side-effects-associated with other T-cell targeting anti-CD3/anti X bi-specifics, with un-protected CD3 binding domain, being developed for cancer immune therapy. Overall, the manuscript is clear, concise and very well written.

General/Specific comments:

1. Using PBMCs as effectors with HELA and Skov-3 tumor cell lines, the authors assessed differences in killing for PROT-TCBs with three linkers in response to cleavage by cellular – as opposed to recombinant – proteases, and show for both cell lines, the combined linker (MMP-2,-9 and matriptase) TCB promotes better killing. A trend for enhanced killing in Hela vs. Skov-3 cells is observed. The authors attribute this difference to cell-specific differences in folate receptor expression, and state that the expression of FOLR1 in Hela cells is ~2 mio antigen binding sites /

cell' vs. '...about 0.1 mio antigen binding sites/ cell'. However, the authors provide no data or references to support this contention, and a brief review of the literature suggests that there is no such difference in FOLR1 expression between Hela and Skov-3 cells; indeed, one recent publication suggests, based on quantitation of protein expression, that the two cell lines are either equivalent in FOLR1 expression or that Skov-3 tend to express slightly higher levels of FOLR1 (PMC5620616, Pharmaceuticals (Basel), September 2017). Indeed, as the authors themselves state on page 5 of this manuscript, the efficacy of TCB-mediated tumor killing is a direct reflection of tumor target expression levels. Thus, the observation that pre-cleaved Prot-FOLR1-TCB mediates equivalent killing of both cell lines at all concentrations is more consistent with an equivalent level of FOLR1 expression on Hela and Skov-3 than their being a ~200-fold difference, and suggests that the difference in unmasking of the antiCD3 fAb may be due to a difference in protease expression level rather than receptor expression, as the authors claim.

2. In figure 6, the authors next assess/compare the killing mediated by: (i) the combined linker PROT-FOLR1-TCB added in un-cleaved form in comparison to the same Prot-FOLR1-TCB pre-cleaved ex vivo with matriptase, (ii) to non-cleavable Prot-FOLR1-TCB, (iii) to a linker-free FOLR1-TCB or (iv) to an untargeted TCB. In Hela cells, equivalent killing is observed for all treatments at all doses, excluding the noncleavable linker, which mediates modest killing at high concentration only in Hela, as observed in figure 5a. In contrast, while Skov-3 cells were equivalent to Hela in response to treatment with the matriptase-cleaved Prot-FOLR1-TCB (which contains a matriptase linker) in figure 5a Skov-3 cells treated here with a linker-free version of the same FOLR1-TCB demonstrate a dramatic reduction in cell death, and killing is equivalent to what is observed with the pre-cleaved Prot-FOLR1-TCB. Interestingly, while uncleaved and cleaved Prot-FOLR1-TCB behave similarly in Hela, in Skov-3 cells the uncleaved Prot-FOLR1-TCB induces significantly less killing. While the authors again attribute this difference to a higher expression of FOLR1 in Hela cells, the observation in figure 5a that FOLR1-TCB, absent linker, mediates equivalent killing of the two cell lines raises the question of why in figure 6a killing mediated by FOLR1-TCB is significantly decreased in Skov-3 cells compared to Hela. In the absence of clear data that there are any differences in folate receptor expression in the two cell lines, a possible reason – and perhaps a consistent explanation for these observations – might be that there are differences in both the type(s) and quantities of proteases produced by Hela and Skov-3 cells. Such an explanation would be consistent with the results of figure 6b, where differences in cytokine production by effector T cells could be readily explained by differences in efficiency of cleavage of the linkers to the scFv masking the antiCD3 fAb, resulting in suboptimal T cell activation.

Main Concern:

My main concern with this manuscript in its present form is that the authors do not provide any in vivo data to support their main contention regarding in vivo site-specific activation of Prot-TCB and enhanced safety. Specifically, I would like to see some data to address the following questions:

- a. In in vivo tumor models what proportion of the anti-idiotype anti-CD3 scFv is cleaved and Prot-TCB activated? In vivo other cell types (eg. epithelial cells) may express matriptase.
- b. Can the activated TCB detected in circulation (circulating levels) and for how long (t_{1/2})
- c. Do activated TCB levels increase in circulation over time? If so, what are the safety implications of this activated TCB in circulation. This final question is critical as this directly relates to the main argument for this intriguing format concept.

As several conditional activation formats have been proposed/described (also referenced by the authors), I believe answers to above questions are important and would make this Prot-TCB concept and manuscript extremely strong and publishable in Nature Communications.

Reviewers' comments:

Reviewer #1 (Remarks to the Author):

The authors describe a method of masking the anti CD3-specificity of a T cell bispecific antibody (TCB) using an anti-idiotypic anti-CD3 domain that interferes with the normal anti-CD3 binding function of the TCB. This technique enabled the authors to “activate” the TCB by using a cleavable linker between the two domains that is cleaved by proteases naturally expressed by tumor cells that express FOLR1. Although the technique of masking antibodies has been previously described (e.g. Tzou et al 2017), this manuscript is the first to describe the technique in detail using a TCB.

I recommend this paper be accepted pending moderate revisions. My main concerns are:

1. Grammar and sentence flow should be improved. Especially paragraphs in the discussion seem disjointed.

The reviewer is right, we changed the disjointed paragraphs.

2. Some figure titles describe the outcome and some merely mention the assay used. Amend titles to convey main message

The reviewer is right, we aligned titles.

3. HrcEpiC cells are not listed in the materials section

The reviewer is right the cells are mentioned now.

4. For most figures, a consistent dose response curve is shown. In figure 7 and 8, only single antibody doses are used. Please explain the rationale for this. A dose response curve provides significantly more information and would benefit readers.

The reviewer is right, however the patient-derived material was limited, that's why we could not titrate the antibodies, but just used a concentration that was in saturation for 2D assays with cell lines.

5. At the end of the conclusion, the sentence about MABEL seems out of place.

The reviewer is right, we deleted this sentence as this is not needed at this time-point.

6. On page 8, the last two paragraphs start with “Table 1)”, which did not make sense to me.

The reviewer is right there was a format problem, that is now corrected.

7. Figure 3B: It is unclear what kind of statistics was done here and whether the provided images are simple representative images.

The reviewer is right and we propose to add a supplemental figure which better transports the statistic relevance of data depicted in Figure 3. It has to be mentioned that the AFM data shown in Figure 3 nicely match observations made with NS-TEM, and that both, AFM as well as NS-TEM data, confirm the structures expected. The underlying samples have been prepared differently, also, the imaging and data analysis process is different. As we achieved convergence within independent data sets, AFM and EM, we actually have a high confidence that the data shown in Fig. 3 are meaningful indeed.

It can be mentioned that observations made on a small ensembles of individual objects are statically meaningful as long the objects have been randomly selected out of a population of more than a billion

unknown objects. The likelihood to pick similarly structured or sized objects out of a macroscopic population is very poor for all heterogeneously composed mixtures.

8. In figure 4C-D: It would help the reader to use the word “unspecific fusion” in the figure description. The reviewer is right, we described the “unspecific fusion” now in the figure description.

9. Throughout the manuscript, different Prot-FOLR1-TCBs are used (with different linkers). Especially in figure 5A, but also in the other figures and in the text throughout the manuscript, it is unclear which linker is used.

The reviewer is right, we added the linker descriptions and a table mentioning all linker sequences.

Reviewer #2 (Remarks to the Author):

In this manuscript the authors describe an interesting T cell Targeting bi-specific format (Prot-TCB) for cancer immune-therapy. The concept of attaching an anti-idiotypic anti-CD3 scFV with a protease sensitive linkers to mask the anti-CD3 binding domain is intriguing. The authors provide excellent and detailed information on arriving at the optimum format design and showing biophysical/biochemical, structural (NS-TEM) and in vitro functional characterization (CD3 domain activation by r-protease, cell lines and fresh tumor potentially expressing MMP 2, 9 and/or matrilysin) for this new “pro-drug” format (Prot-TCB). They also provide data for in vitro serum stability and preliminary manufacturability for Prot-TCB format. The authors suggest that the main advantage of such a “pro-drug” format will be tumor site specific activation of such molecules that will potentially enhance safety and reduce side-effects-associated with other T-cell targeting anti-CD3/anti X bi-specifics, with un-protected CD3 binding domain, being developed for cancer immune therapy. Overall, the manuscript is clear, concise and very well written.

General/Specific comments:

1. Using PBMCs as effectors with HELA and Skov-3 tumor cell lines, the authors assessed differences in killing for PROT-TCBs with three linkers in response to cleavage by cellular – as opposed to recombinant – proteases, and show for both cell lines, the combined linker (MMP-2,-9 and matrilysin) TCB promotes better killing. A trend for enhanced killing in HeLa vs. Skov-3 cells is observed. The authors attribute this difference to cell-specific differences in folate receptor expression, and state that the expression of FOLR1 in HeLa cells is ‘2 mio antigen binding sites / cell’ vs. ‘...**about 0.1 mio antigen binding sites/ cell**’. **However, the authors provide no data or references to support this contention**, and a brief review of the literature suggests that there is no such difference in FOLR1 expression between HeLa and Skov-3 cells; indeed, one recent publication suggests, based on quantitation of protein expression, that the two cell lines are either equivalent in FOLR1 expression or that Skov-3 tend to express slightly higher levels of FOLR1 (PMC5620616, Pharmaceuticals (Basel), September 2017). Indeed, as the authors themselves state on page 5 of this manuscript, the efficacy of TCB-mediated tumor killing is a direct reflection of tumor target expression levels. Thus, the observation that pre-cleaved Prot-FOLR1-TCB mediates equivalent killing of both cell lines at all concentrations is more consistent with an equivalent level of FOLR1 expression on HeLa and Skov-3 than their being a ~200-fold difference, and suggests that the difference in unmasking of the antiCD3 fAb may be due to a difference in protease expression level rather than receptor expression, as the authors claim.

The reviewer is right, we did not show the data for the FOLR1 expression levels. We now added these data that support the suggestion that HeLa cells in our hands do express more FOLR1 compared to Skov-3 cells. The efficacy of TCB-mediated tumor killing is a direct reflection of tumor target expression levels. The reviewer is partially right, the pre-cleaved Prot-FOLR1-TCB mediates more efficient killing of HeLa cells regarding EC50 values, whereas the killing at saturating concentrations of the activated Prot-FOLR1-TCB is comparable for both cell lines.

2. In figure 6, the authors next assess/compare the killing mediated by: (i) the combined linker PROT-FOLR1-TCB added in un-cleaved form in comparison to the same Prot-FOLR1-TCB pre-cleaved ex vivo with matriptase, (ii) to non-cleavable Prot-FOLR1-TCB, (iii) to a linker-free FOLR1-TCB or (iv) to an untargeted TCB. In HeLa cells, equivalent killing is observed for all treatments at all doses, excluding the noncleavable linker, which mediates modest killing at high concentration only in HeLa, as observed in figure 5a. In contrast, while Skov-3 cells were equivalent to HeLa in response to treatment with the matriptase-cleaved Prot-FOLR1-TCB (which contains a matriptase linker) in figure 5a Skov-3 cells treated here with a linker-free version of the same FOLR1-TCB demonstrate a dramatic reduction in cell death, and killing is equivalent to what is observed with the pre-cleaved Prot-FOLR1-TCB. Interestingly, while un-cleaved and cleaved Prot-FOLR1-TCB behave similarly in HeLa, in Skov-3 cells the un-cleaved Prot-FOLR1-TCB induces significantly less killing. While the authors again attribute this difference to a higher expression of FOLR1 in HeLa cells, the observation in figure 5a that FOLR1-TCB, absent linker, mediates equivalent killing of the two cell lines raises the question of why in figure 6a killing mediated by FOLR1-TCB is significantly decreased in Skov-3 cells compared to HeLa. In the absence of clear data that there are any differences in folate receptor expression in the two cell lines, a possible reason – and perhaps a consistent explanation for these observations – might be that there are differences in both the type(s) and quantities of proteases produced by HeLa and Skov-3 cells. Such an explanation would be consistent with the results of figure 6b, where differences in cytokine production by effector T cells could be readily explained by differences in efficiency of cleavage of the linkers to the scFv masking the antiCD3 fAb, resulting in suboptimal T cell activation.

The reviewer is right, we added the data that support the suggestion that HeLa cells do express more FOLR1 compared to Skov-3 cells. The efficacy of TCB-mediated tumor killing is a direct reflection of tumor target expression levels. The reviewer is right, the un-cleaved Prot-FOLR1-TCB mediates less efficient killing of Skov-3 cells compared to the pre-cleaved Prot-FOLR1-TCB and the FOLR1-TCB. This could indeed be due to different MMP9 or matriptase expression levels. In the tumor microenvironment different cell types (tumor cells, fibroblasts and immune cells) are described to produce MMP9 and matriptase.

Main Concern:

My main concern with this manuscript in its present form is that the authors do not provide any in vivo data to support their main contention regarding in vivo site-specific activation of Prot-TCB and enhanced safety.

Specifically, I would like to see some data to address the following questions:

a. In in vivo tumor models what proportion of the anti-idiotype anti-CD3 scFv is cleaved and Prot-TCB activated? In vivo other cell types (eg. epithelial cells) may express matriptase.

b. Can the activated TCB detected in circulation (circulating levels) and for how long (t1/2)

c. Do activated TCB levels increase in circulation over time? If so, what are the safety implications of this activated TCB in circulation. This final question is critical as this directly relates to the main argument for this intriguing format concept.

As several conditional activation formats have been proposed/described (also referenced by the authors), I believe answers to above questions are important and would make this Prot-TCB concept and manuscript extremely strong and publishable in Nature Communications.

The reviewer is right, we now added the full in vivo data package including stability of pro-TCB in non-tumor bearing mice as well as efficacy data. In addition we added in vivo data, showing that we have no hint, that Prot-FOLR1-TCB that is activated in the tumor does accumulate in serum (tumor leakage of activated Prot-FOLR1-TCB).

a. We work on an assay to quantify the amount of activated pro-TCB in the tumor, however from the efficacy study we clearly see a difference for Prot-FOLR1-TCBs with two different cleavage sites suggesting that one is cleaved more efficiently than the other. Furthermore the formation of active TCB (i.e. the proportion of cleaved anti-idiotype anti-CD3 scFv and activated Prot-FOLR1-TCB) was estimated from the bioavailability of the active TCB after administration of the Prot-TCBs, both in tumor bearing and non-tumor bearing mice. This information is added in the manuscript now (Figure 7, result section "Prot-FOLR1-TCB is activated in vivo and no tumor leakage of activated Prot-FOLR1-TCB could be detected in serum").

b. Can the activated TCB detected in circulation (circulating levels) and for how long ($t_{1/2}$)

Yes, we developed an ELISA to quantify the bioavailability of active Prot-FOLR1-TCB in circulation (serum). We could show that there is ~5% of active Prot-FOLR1-TCB in circulation independent of tumor as it was comparable in non-tumor bearing and in tumor-bearing mice. The activated Prot-FOLR1-TCBs were followed in circulation up to 7 days. Values were used for calculation of AUC 0-168 h for calculation of bioavailability of the active TCB. These data did not allow calculation of a half-life (see also answer to c. regarding potential accumulation).

c. The potential accumulation of activated TCB following administration of prot-TCB was assessed following weekly administration over 4 weeks in the xenograft model. There was only slight increase in trough concentrations of active TCB comparing trough levels after 1st and 4th dose (data not shown). The increase in trough levels was similar to the one observed after weekly dosing of the activated TCB itself. Overall, there is no indication for an undue accumulation of activated TCB levels over time.

Reviewers' Comments:

Reviewer #1:

Remarks to the Author:

Comments to author

The authors have satisfactorily addressed all previous comments.

Major comments

None

Minor comments

none

Reviewer #2:

Remarks to the Author:

I had reviewed this manuscript earlier and had submitted my detailed feedback/concerns.

Now, I have reviewed the revised manuscript and the specific revisions made to the manuscript.

The authors have very nicely addressed all my concerns, both general and specific. The addition of data (Fig 7) on the in vivo characterization/ stability of the Prot-FOLR1-TCBs (including linkers) makes this manuscript very strong. Overall, this manuscript will add new information to the field and help in the design of novel and hopefully better therapies. I would recommend this revised manuscript be published.